# Incorporating Interventional Independence Improves Robustness against Interventional Distribution Shift

## Abstract

We consider learning discriminative representations of variables related to each other via a causal graph. To learn representations that are robust against interventional distribution shifts, the training dataset is augmented with interventional data in addition to existing observational data. However, even when the underlying causal model is known, existing approaches treat interventional data like observational data, ignoring the independence relations resulting from these interventions. This leads to representations that exhibit large disparities in predictive performance on observational and interventional data. The performance disparity worsens when the quantity of interventional data available for training is limited. In this paper, (1) we first identify a strong correlation between this performance disparity and adherence of the representations to the statistical independence conditions induced by the underlying causal model during interventions. (2) For linear models, we derive sufficient conditions on the proportion of interventional data during training, for which enforcing statistical independence between representations corresponding to the intervened node and its non-descendants during interventions can lower the test-time error on interventional data. Following these insights, we propose RepLIn, an algorithm to explicitly enforce this statistical independence during interventions. We demonstrate the utility of RepLIn on synthetic and real face image datasets. Our experiments show that RepLIn is scalable with the number of nodes in the causal graph and is suitable to improve the robustness of representations against interventional distribution shifts of both continuous and discrete latent variables compared to the ERM baselines.

## 1 Introduction

We consider the problem of learning discriminative representations corresponding to latent random variables for their prediction from their observable data. The relationship between these latent variables can be modeled using directed acyclic graphs (DAGs) called *causal graphs*. These latent variables usually correspond to semantic concepts such as the color of an object, the level of glucose in the blood, and a person's age. Causal modeling allows manually altering the causal graph and observing its effects on the data, for instance, by consuming an insulin inhibitor and measuring the glucose level in the blood. This procedure is known as a *causal intervention*, and the data collected through this procedure is called *interventional data*. In contrast, data collected without intervention is known as *observational data*. Several types of interventions are possible on a causal graph, of which we are interested in *hard interventions* where we manually set the value of one or more variables. Intervening on a node renders it statistically independent of its *parent nodes* in the causal graph[1]. See (Peters et al., 2017, Chapter 6) and (Pearl, 2009, Chapter 3).

Suppose the latent variables are $A$ and $B$, such that $A \rightarrow B$ ($A$ causes $B$) during observations. An attribute-specific representation $\boldsymbol{F}_A$ corresponding to $A$ learned by a model from observational training data alone may contain information about its child node $B$ due to the association between $A$ and $B$. Subsequently, these models show a performance drop on data collected through intervention on $B$ during inference. In

---

[1] For ease of use, we refer to "statistical independence" as "independence", and "hard interventions" as "interventions". We will also use "features" to describe the representations a model learns from data.

other words, these models are not robust against *interventional distribution shifts*. Interventional data samples are included in the training data to learn models that are robust to interventional distribution shifts. For example, in (Sauer & Geiger, 2021; Gao et al., 2023), interventional data was generated using data augmentations to train image classification models invariant to texture and background. In some works such as (Arjovsky et al., 2019; Heinze-Deml & Meinshausen, 2021), interventional data is treated merely as data sourced from different domains or *environments*, and they do not consider the *explicit* statistical independence relations that arise from interventions[2]. As we demonstrate, ignoring these independence relations may result in representations still susceptible to interventional distribution shifts during inference. Additionally, performing interventions is often challenging, thus limiting the amount of interventional data available for training. Therefore, causally motivated learning is necessary to improve the robustness of learned representations against interventional distribution shifts.

We first consider a simple case study in which we observe that models that do not learn independent representations during interventions show a performance drop on interventional data. We then derive sufficient conditions on the proportion of interventional data during training, for which enforcing linear independence between interventional features of linear models during training can reduce test-time error on interventional data. Following the theoretical assessment, we propose "Representation Learning from Interventional Data" (**RepLIn**), an algorithm to train models with improved robustness against interventional distribution shifts. We confirm the utility of RepLIn on synthetic (Sec. 5.1) and real face image datasets (Sec. 5.2) and demonstrate its scalability to the number of nodes (Sec. 6.2).

To summarize our contributions,

- We demonstrate a positive correlation between accuracy drop during interventional distribution shift and dependence between representations corresponding to the label node and its children. We refer to this as "interventional feature dependence" (Sec. 3.3).

- We theoretically explain why linear ERM models are susceptible to interventional distribution shifts in the regime of linear causal models. In the same setting, we theoretically and empirically show that enforcing linear independence between interventional features improves robustness when sufficient interventional data is available during training and establish the sufficient condition (Sec. 3.4).

- We propose a novel training algorithm that combines these insights and demonstrates that this model minimizes the drop in accuracy under interventional distribution shifts by explicitly enforcing independence between interventional features (Sec. 4).

## 2 Related Works

**Identifiable Causal Representation Learning (ICRL)** (Locatello et al., 2019; Schölkopf et al., 2021; Hyvärinen et al., 2024) seeks to learn representations of the underlying causal model under certain assumptions (Hyvärinen et al., 2024), and is, therefore, important to interpretable representation learning. However, we are interested in a broader class of discriminative representation learning when some underlying causal relations are known. In contrast to learning the entire causal model, we seek to exploit the known independence relations from interventions to learn discriminative representations that are robust against these interventions. We provide a detailed review of ICRL in App. C.

**Interventional data** is key in causal discovery (Eberhardt et al., 2005; Yu et al., 2019; Ke et al., 2019; Lippe et al., 2022a; Wang et al., 2022) as one can only retrieve causal relations up to Markov equivalent graph without interventions or assumptions on the causal model. For example, known interventional targets have been used for unsupervised causal discovery of linear causal models (Subramanian et al., 2022), interventional and observational data have been leveraged for training a supervised model for causal discovery (Ke et al., 2022), and interventions with unknown targets were used for differentiable causal discovery (Brouillard et al., 2020). Interventional data also find applications in reinforcement learning (Gasse et al., 2021; Ding et al., 2022) and recommendation systems (Zhang et al., 2021; Krauth et al., 2022; Luo et al., 2024). While this

---

[2]Note that distribution shifts due to differing environments is more general than interventional distribution shift. However, in this work, we argue against an agnostic treatment against interventional distribution shift.

body of work focuses on discovering causal relations in the data, our work considers how to leverage known causal relations to learn data representations that are robust to distribution shifts induced by interventions.

**Training with group-imbalanced data** leads to models that suffer from group-bias during inference. In such cases, resampling the data according to the inverse sample frequency can improve generalization and robustness. Studies such as (Gulrajani & Lopez-Paz, 2021; Idrissi et al., 2022) have shown that ERM with resampling is effective against spurious correlations and is a strong baseline for domain generalization. Recent work such as dynamic importance reweighting (Fang et al., 2020), SRDO (Shen et al., 2020), and MAPLE (Zhou et al., 2022) *learn to resample* using a separate validation set that acts as a proxy for the test set. However, learning such a resampling requires a large dataset of both *observational* and *interventional* data, which is often not practically feasible. In contrast, we will exploit known independence relations during interventions to improve robustness to interventional distributional shifts.

## 3 The Learning from Interventional Data Problem

**Notation:** Random variables and random vectors are denoted by regular (e.g., $A$) and bold (e.g., $\boldsymbol{a}$) sans-serif uppercase characters, respectively. The distribution of a random variable $A$ is denoted by $P_A$.

We now formally define the problem of interest in this paper, namely *learning discriminative representations to predict latent variables that are robust against interventional distribution shifts*[3], in general terms, and examine a specific case study in Sec. 3.1. The learning problem is characterized by a DAG $\mathcal{G}$ that causally relates our attributes of interest $A_1, \ldots, A_m$, and $B$. Let $\mathbf{Pa}_B = \{A_1, \ldots, A_m\}$ denote the parents of the attribute $B$. These attributes along with other unobserved exogenous variables $\boldsymbol{U}$, generate the observable data $\boldsymbol{X}$, i.e., $\boldsymbol{X} = g_{\boldsymbol{X}}(B, A_1, \ldots, A_m, \boldsymbol{U})$.

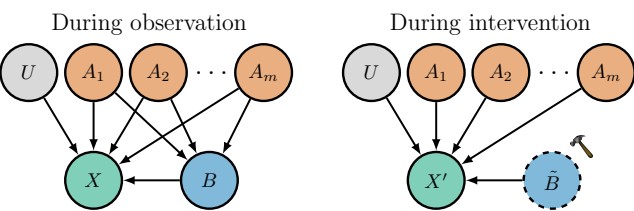

Figure 1: **Causal graph modification due to intervention:** During observation, $B$ is the effect of its parent variables $\mathbf{Pa}_B = \{A_1, \ldots, A_m\}$. When we intervene on $B$, it becomes statistically independent of its parents.

During interventions, the variable $B$ is set to values drawn from a known distribution independent of $\mathbf{Pa}_B$. Therefore, the post-intervention variable $B$ (denoted by $\tilde{B}$) is statistically independent of its parents, i.e., $\tilde{B} \perp\!\!\!\perp \mathbf{Pa}_B$, as shown in Fig. 1. Although $g_{\boldsymbol{X}}$ is not affected by this intervention, the distribution of $\boldsymbol{X}$ (now denoted by $\boldsymbol{X}'$) will change since it is a function of $B$. Note that to learn representations that are robust against distribution shift due to intervention on $B$, our setting does not provide the knowledge of any node apart from $B$ and its parents in this causal graph or of any causal relations between $A_1, \ldots, A_m$. We also do not have restrictions on the functional form of causal relations between $A_1, \ldots, A_m, B$, and $\boldsymbol{X}$, or on their marginal distributions. For training, data samples from both observational and interventional distributions are available, i.e., $\mathcal{D}^{\text{train}} = \mathcal{D}^{\text{obs}} \cup \mathcal{D}^{\text{int}}$ where $\mathcal{D}^{\text{obs}} \sim P(\boldsymbol{X}, B, A_1, \ldots, A_m)$ and $\mathcal{D}^{\text{int}} \sim P(\boldsymbol{X}', \tilde{B}, A_1, \ldots, A_m)$. Given $\mathcal{D}^{\text{train}}$ and $\mathcal{G}$, the goal is to learn attribute-specific discriminative representations $\boldsymbol{F}_B = h_B(\boldsymbol{X})$ and $\boldsymbol{F}_{A_i} = h_{A_i}(\boldsymbol{X})$ that are robust against distribution shifts due to intervention on $B$.

### 3.1 Does Accuracy Drop during Interventions Correlate with Interventional Feature Dependence?

First, we consider a motivating case study on a synthetic dataset and establish a correlation between the accuracy drop on interventional data and statistical dependence between the attribute representations under intervention. We will then estimate the strength of this correlation and theoretically investigate whether this correlation can be exploited to improve the robustness against interventional distribution shifts.

**Problem Setting:** Consider the causal graph shown in Fig. 2a. Here, $A$ and $B$ are binary random variables that generate the observed data $\boldsymbol{X} \in \mathbb{R}^2$. $\boldsymbol{X}$ is also affected by an unobserved noise variable $\boldsymbol{U}$. Therefore, functionally $\boldsymbol{X} = g_{\boldsymbol{X}}(A, B, \boldsymbol{U})$. $A$ itself could be a function of unobserved random factors that are of no predictive interest to us. Therefore, we model $A \sim \text{Bernoulli}(0.6)$. The distribution of $B$ is only affected by

---

[3]We use "discriminative" to explicitly state that the purpose of these representations is robust prediction and not data generation. Information loss with improved robustness is therefore acceptable.

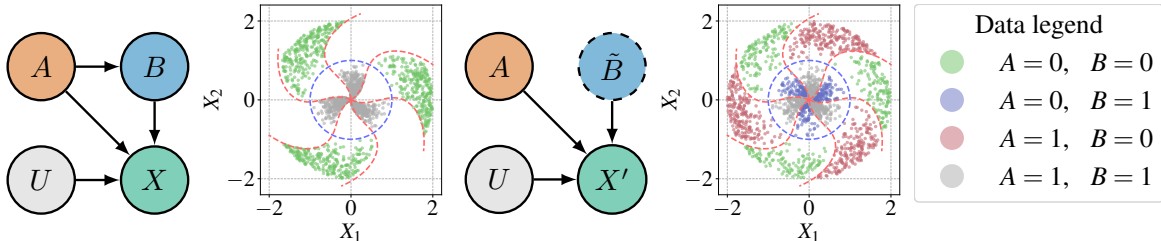

(a) Observational graph and data  (b) Interventional graph and data

Figure 2: **An illustration of Windmill Dataset:** $A$ and $B$ are binary random variables that are causally linked to each other and $\boldsymbol{X}$, as shown in (a). By intervening on $B$ as shown in (b), we make $A \perp\!\!\!\perp \tilde{B}$. $\boldsymbol{X} = g_{\boldsymbol{X}}(A, B, \boldsymbol{U})$ where $\boldsymbol{U}$ denotes unobserved noise variables. The true decision boundaries for predicting $A$ and $B$ from $\boldsymbol{X}$ are shown in red and blue dashed lines, respectively. See App. F for a detailed description.

$A$, as denoted by the arrow between them. Analytically, $B := A$, where $:=$ indicates the causal assignment operator, following (Peters et al., 2017). Visually, the observed data looks like a windmill. The value of $A$ determines the windmill's blade, and $B$ determines the radial distance. We shear the windmill blades according to a sinusoidal function of the radial distance. To make the data more stochastic, the points' precise angle and radial distance are sampled from an unobserved distribution independent of $A$ and $B$. In Fig. 2b, we intervene on $B$, modeled as $\tilde{B} \sim \text{Bernoulli}(0.5)$. This induces a change in the distribution of $B$ and subsequently that of $\boldsymbol{X}$. Since the intervention is independent of $A$, $\tilde{B}$ is also independent of $A$, denoted by removing the arrow between $A$ and $\tilde{B}$. Note that $g_{\boldsymbol{X}}$ is unaffected by this intervention. The exact mathematical formulation of the data-generating process is provided in App. F.

**Learning task:** The task is to accurately predict $A$ and $B$ from $\boldsymbol{X}$ at test time. We have $N$ samples for training, where $\beta N$ are interventional and $(1-\beta)N$ are observational with $0 < \beta < 1$ typically being a small value. For this demonstration, we set $N = 40,000$, $\beta = 0.01$. Therefore, we have 39,600 observational and 400 interventional samples. We train a feed-forward network with two hidden layers to learn representations $\boldsymbol{F}_A$ and $\boldsymbol{F}_B$ corresponding to $A$ and $B$, respectively. We normalize them by dividing each by their corresponding $L_2$ norm. Separate linear classifiers predict $A$ and $B$ from $\boldsymbol{F}_A$ and $\boldsymbol{F}_B$ respectively. By construction, $g_{\boldsymbol{X}}$ in the data-generating process is a one-to-one mapping. Therefore, predicting $A$ and $B$ from $\boldsymbol{X}$ accurately is possible. However, the true decision boundary for $A$ is more complex than that of $B$[4]. Therefore, the model may rely on information from $B$ to predict $A$ due to their association during observation, similar to the concept of simplicity bias from (Shah et al., 2020). As a result, $\boldsymbol{F}_A$ may contain information about $B$ even during interventions when $A \perp\!\!\!\perp B$.

| ERM version | Accuracy in predicting $A$ | | | Accuracy in predicting $B$ | | | NHSIC |
|---|---|---|---|---|---|---|---|
| | Observation | Intervention | Relative drop | Observation | Intervention | Relative drop | |
| Vanilla | 99.97 | 58.56 | 0.414 | 100 | 100 | 0 | 0.786 |
| w/ Resampling | 93.24 | 68.65 | 0.264 | 100 | 99.99 | $10^{-4}$ | 0.537 |

Table 1: The relative drop in accuracy in predicting $A$ correlates well with a gap in the measure of dependence between the learned representations on interventional data.

**Observations:** Following the standard ERM framework, the cross entropy errors in predicting $A$ and $B$ from $\boldsymbol{F}_A$ and $\boldsymbol{F}_B$, respectively, provide the training signal. The statistical loss function can be written as $\mathcal{L}_{\text{total}}(f, \boldsymbol{X}) = \mathbb{E}_{P_{\text{train}}}[\mathcal{L}_{\text{pred}}(f, \boldsymbol{X})]$. The training distribution is a mixture of observational and interventional distributions with $(1 - \beta)$ and $\beta$ acting as the corresponding mixture weights. Thus, $\mathcal{L}_{\text{total}}(f, \boldsymbol{X}) = (1-\beta)\mathbb{E}_{P_{\text{obs}}}[\mathcal{L}_{\text{pred}}(f, \boldsymbol{X}^{\text{obs}})] + \beta\mathbb{E}_{P_{\text{int}}}[\mathcal{L}_{\text{pred}}(f, \boldsymbol{X}^{\text{int}})]$. Tab. 1 shows the accuracy of ERM in predicting $A$ and $B$ on observational and interventional data during validation. Ideally, *we expect no drop in accuracy from observation to intervention* if the learned representations are robust against interventional dis-

---

[4]We informally define "complexity" as the minimum polynomial degree required to approximate the decision boundary.

tribution shift. However, we observe that ERM performs only slightly better than random chance in predicting $A$ on interventional data. As a remedy, we consider a stronger version of ERM by sampling observational and interventional data in separate batches. This is equivalent to sampling interventional data $\left(\frac{1-\beta}{\beta}\right)$-times as observational data. Therefore, we refer to this version as "ERM-Resampled". The equivalent loss for a learning function $f$ in ERM-Resampled is $\mathcal{L}_{\text{total}}(f, \boldsymbol{X}) = \mathbb{E}_{P_{\text{obs}}}\left[\mathcal{L}_{\text{pred}}(f, \boldsymbol{X}^{\text{obs}})\right] + \mathbb{E}_{P_{\text{int}}}\left[\mathcal{L}_{\text{pred}}(f, \boldsymbol{X}^{\text{int}})\right]$. Note that $\beta$ does not appear in $\mathcal{L}_{\text{total}}(f, \boldsymbol{X})$ due to resampling. Although ERM-Resampled performs better than vanilla ERM, we observe that ERM-Resampled still exhibits a large drop in predictive accuracy between observational and interventional data during inference. Also, observe the drop in observational accuracy of ERM-Resampled in predicting $A$ as it improved interventional accuracy. As we will show in Sec. 3.4, the reduced observational accuracy is due to the removal of spurious information that had previously improved its observational accuracy.

## 3.2 Measuring Statistical Dependence Between Interventional Features

A key characteristic of hard interventions in causal graphs is that the variable being intervened upon becomes independent of all its nondescendants. Since the predictive accuracy on the parent node is affected by intervention, we hypothesize that *the representation corresponding to the parent node is dependent on the child node during intervention.* Therefore, to verify our hypothesis, we measure the dependence between the representations. We measure the dependence between the representations instead of between the representations and the latent attributes because we aim to learn robust representations for every attribute.

**Dependence Measure:** To measure dependence between a pair of high-dimensional continuous random variables $\boldsymbol{X}$ and $\boldsymbol{Y}$, we use HSIC (Gretton et al., 2005), a non-parametric measure of dependence. Given $N$ i.i.d. samples $\mathcal{X} = \left\{\boldsymbol{x}^{(i)}\right\}_{i=1}^{N}$ and $\mathcal{Y} = \left\{\boldsymbol{y}^{(i)}\right\}_{i=1}^{N}$ from $\boldsymbol{X}$ and $\boldsymbol{Y}$, empirical HSIC between these samples can be computed as $\text{HSIC}(\mathcal{X}, \mathcal{Y}) = \frac{1}{(N-1)^2}\text{Trace}\left[\boldsymbol{K}_X \boldsymbol{H} \boldsymbol{K}_Y \boldsymbol{H}\right]$, where $\boldsymbol{H}$ is the $N \times N$ centering matrix, and $\boldsymbol{K}_X, \boldsymbol{K}_Y \in \mathbb{R}^{N \times N}$ are Gram matrices whose $(i,j)^{\text{th}}$ entries are $k_X\left(\boldsymbol{x}^{(i)}, \boldsymbol{x}^{(j)}\right)$ and $k_Y\left(\boldsymbol{y}^{(i)}, \boldsymbol{y}^{(j)}\right)$, respectively. Here, $k_X$ and $k_Y$ are the kernel functions associated with a universal kernel (e.g., RBF kernel). Since HSIC is unbounded, we normalize it as $\text{NHSIC}(\mathcal{X}, \mathcal{Y}) = \frac{\text{HSIC}(\mathcal{X}, \mathcal{Y})}{\sqrt{\text{HSIC}(\mathcal{X},\mathcal{X})\,\text{HSIC}(\mathcal{Y},\mathcal{Y})}}$, following (Cortes et al., 2012; Cristianini et al., 2001).

We use the NHSIC metric to compare the statistical dependence between the features in the WINDMILL problem. Tab. 1 shows the difference in NHSIC values between the features $F_A$ and $F_B$ from interventional data. We observe that ERM-Resampled learns features with less statistical dependence during interventions than vanilla ERM. A larger interventional feature dependence indicates a larger violation of the underlying statistical independence relations that result from interventions in the causal graph.

## 3.3 Strength of Correlation between Drop in Accuracy and Interventional Features Dependence

How strong is the observed correlation between the dependence of features and the drop in accuracy? For a given combination of predictive task and dataset, does it hold for a variety of hyperparameter settings? To answer these questions, we train several models under the ERM-Resampled setting described in Sec. 3.1. To learn representations, we use feed-forward networks, each with one to six hidden layers and with 20 to 200 hidden units. We also randomly set the number of training epochs to use early-stopping as a regularizer, as described in (Sagawa et al., 2020). To measure the robustness of a model to interventional distribution shift, we evaluate the relative drop in accuracy between observational and interventional data: $\text{Rel}.\Delta = \frac{\text{Obs acc.} - \text{Int acc.}}{\text{Obs acc.}}$.

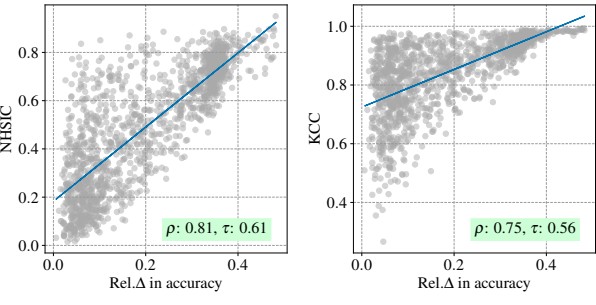

(a) Rel.$\Delta$ against NHSIC    (b) Rel.$\Delta$ against KCC

Figure 3: Across models with different capacities, a relative drop in accuracy is always accompanied by interventional feature dependence, while the corollary does not hold. Interventional feature dependence is measured using NHSIC and KCC.

In Fig. 3, we plot the relative drop in accuracy
against the interventional feature dependence. In
addition to NHSIC, we also use kernel canonical correlation (KCC) (Bach & Jordan, 2002) to measure the
dependence. We observe that all models with a high relative drop in accuracy also have a large interventional feature dependence (see top-right regions of the plots). However, the corollary is not true – a large
interventional feature dependence does not mean a relative drop in accuracy. Therefore, we conclude that
*a relative drop in accuracy is always accompanied by interventional feature dependence.* The strength of the
correlation between the relative drop in accuracy and interventional feature dependence is quantitatively
measured using Spearman rank correlation coefficient ($\rho$) (Spearman, 1904) and Kendall rank correlation
coefficient ($\tau$) (Kendall, 1938). In Fig. 3a, $\rho = 0.81$ and $\tau = 0.61$ when the dependence is measured using
NHSIC, indicating that the correlation we noted in Sec. 3.2 can be observed for a wide range of models.
When KCC is used for measuring dependence between interventional representations, $\rho = 0.75$ and $\tau = 0.56$
as shown in Fig. 3b. Note that the correlation measures are affected by the choice of measure of dependence.
Both NHSIC and KCC satisfy the postulates for an appropriate measure of dependence in (Rényi, 1959) and
measure dependence from the spectrum of the cross-covariance operator between RKHSs. However, NHSIC
measures the Hilbert-Schmidt norm of the cross-covariance operator while KCC measures its spectral norm.
Therefore, KCC is a "harsher" measure of dependence compared to NHSIC. ~~A similarly strong correlation
can be observed between the relative drop in accuracy and KCC in Fig. 3b.~~

### 3.4 Will Minimizing Dependence between Interventional Features Improve Robustness?

Fig. 3 showed that strong interventional feature dependence always accompanies a large relative drop in
accuracy. Based on this correlation, we may ask the following question: *will minimizing interventional
feature dependence improve the robustness to interventional distribution shifts?* We consider a linear causal
model to answer this question theoretically. The detailed proof of each step is provided in App. B.

**Causal Model:** We use the causal model shown in Fig. 2a with $A$ and $B$ being continuous random variables.
$A$ and $B$ are causally related during observation as $B := w_{AB}A$. The observed data signal $\boldsymbol{X}$ is generated
from $A$ and $B$ as $\boldsymbol{X} := \begin{bmatrix} X_A \\ X_B \end{bmatrix} + \boldsymbol{U}$, where $X_A := w_A A$ and $X_B := w_B B$. $\boldsymbol{U} := \begin{bmatrix} U_A \\ U_B \end{bmatrix}$ is exogenous noise.
$U_A$ and $U_B$ are independent of $A$ and $B$ respectively. We intervene on $B$ as shown in Fig. 2b, severing the
causal relation between $A$ and $B$. The intervened variable is denoted as $B'$ and $B' \perp\!\!\!\perp A$.

**Learning model:** Similar to the case study, the task is to predict the latent variables $A$ and $B$ from observed
data signal $\boldsymbol{X}$. The training dataset is sampled from a training distribution $P_{\text{train}}$ that contains observational
and interventional samples. We model $P_{\text{train}}$ as a mixture of observation distribution $P_{\text{obs}}$ and interventional
distribution $P_{\text{int}}$ with $(1 - \beta)$ and $\beta$ acting as the mixture weights, i.e., $P_{\text{train}} = (1 - \beta)P_{\text{obs}} + \beta P_{\text{int}}$. We
use linear models to learn attribute-specific representations $\boldsymbol{F}_A$ and $\boldsymbol{F}_B$, from which predictions $\hat{A}$ and $\hat{B}$,
respectively, are made using the classifiers. The linear models are parameterized by $\boldsymbol{\Theta}^{(A)}$ and $\boldsymbol{\Theta}^{(B)}$, and the
classifiers are parameterized by $\boldsymbol{c}^{(A)}$ and $\boldsymbol{c}^{(B)}$.

**Statistical Risk:** The parameter matrix of the linear feature extractor described before can be written in
terms of its constituent parameter vectors as $\boldsymbol{\Theta}^{(A)} = \begin{bmatrix} \boldsymbol{\theta}_A^{(A)\top} \\ \boldsymbol{\theta}_B^{(A)\top} \end{bmatrix}$. Assuming zero mean for all latent variables,
the statistical squared error of an arbitrary model in predicting $A$ from an interventional test sample $\boldsymbol{X}$ is,

$$E_A = \underbrace{\left(1 - w_A \boldsymbol{c}^{(A)\top} \boldsymbol{\theta}_A^{(A)}\right)^2 \rho_A^2 + \left(\boldsymbol{c}^{(A)\top} \boldsymbol{\theta}_A^{(A)}\right)^2 \rho_{\boldsymbol{U}_A}^2}_{E_A^{(1)}} + \underbrace{\left(w_B \boldsymbol{c}^{(A)\top} \boldsymbol{\theta}_B^{(A)}\right)^2 \rho_{B'}^2 + \left(\boldsymbol{c}^{(A)\top} \boldsymbol{\theta}_B^{(A)}\right)^2 \rho_{\boldsymbol{U}_B}^2}_{E_A^{(2)}} \quad (1)$$

where $\rho_A^2 = \mathbb{E}_{P_{\text{int}}}\left[A^2\right]$, $\rho_{B'}^2 = \mathbb{E}_{P_{\text{int}}}\left[B'^2\right]$, $\rho_{\boldsymbol{U}_A}^2 = \mathbb{E}_{P_{\text{int}}}\left[U_A^2\right]$, and $\rho_{\boldsymbol{U}_B}^2 = \mathbb{E}_{P_{\text{int}}}\left[U_B^2\right]$. The statistical risk can
be split into two components: (1) $E_A^{(1)}$ in terms of $A$ and $U_A$, and (2) $E_A^{(2)}$ in terms of $B$ and $U_B$. $E_A^{(2)} \neq 0$
when $\boldsymbol{\theta}_B^{(A)} \neq \boldsymbol{0}$. A non-zero $\boldsymbol{\theta}_B^{(A)}$ indicates that the representation $\boldsymbol{F}_A$ is a function of $X_B$, i.e., it learns
a spurious correlation with $B$. Thus the prediction $\hat{A}$ is susceptible to interventions on $B$. In contrast, a
robust model will have $\boldsymbol{\theta}_B^{(A)} = \boldsymbol{0}$, and thus $E_A^{(2)} = 0$. Derivation of Eq. (1) is provided in App. B.1.

**Optimal ERM model:** The optimal ERM model can be obtained by minimizing the expected risk in predicting the latent attributes. Since parameters are not shared between the prediction of $\boldsymbol{a}$ and $\boldsymbol{b}$, we can consider their optimization separately. We will consider the optimization of parameters for predicting $\boldsymbol{a}$ since we are interested in the performance drop in predicting $A$ from interventional data.

$$\boldsymbol{\Theta}^{(A)*}, \boldsymbol{c}^{(A)*} = \underset{\boldsymbol{\Theta}^{(A)}, \boldsymbol{c}^{(A)}}{\operatorname{argmin}} \, \mathbb{E}_{P_{\text{train}}} \left[ \left( A - \boldsymbol{c}^{(A)\top} \boldsymbol{\Theta}^{(A)\top} \boldsymbol{X} \right)^2 \right] \tag{2}$$

For a given training error, there is no unique solution for $\boldsymbol{\Theta}^{(A)}$ and $\boldsymbol{c}^{(A)}$. Therefore, we can equivalently optimize for $\boldsymbol{\psi}_A = \boldsymbol{c}^{(A)\top} \boldsymbol{\Theta}^{(A)\top}$. We can write $\boldsymbol{\psi}_A = \begin{bmatrix} \psi_1 \\ \psi_2 \end{bmatrix}$ where $\psi_1 = \boldsymbol{c}^{(A)\top} \boldsymbol{\theta}_A^{(A)}$ and $\psi_2 = \boldsymbol{c}^{(A)\top} \boldsymbol{\theta}_B^{(A)}$. The learning objective in Eq. (2) then reduces to,

$$\boldsymbol{\psi}_A^* = \underset{\boldsymbol{\psi}_A}{\operatorname{argmin}} \, \mathbb{E}_{P_{\text{train}}} \left[ (A - \boldsymbol{\psi}_A \boldsymbol{X})^2 \right] \tag{3}$$

We can solve Eq. (3) by setting the gradients to zero. To check the robustness of the optimal ERM model, we can verify whether $\psi_2^* = 0$ or not since a robust model will have $\boldsymbol{\theta}_B^{(A)} = \boldsymbol{0}$. Solving Eq. (3), we get:

$$\psi_2^* = \frac{-(1 - \beta) w_B w_{AB} \sigma_A^2 \sigma_{U_A}^2}{T} \neq 0 \tag{4}$$

where $T$ is a non-zero scalar. This implies that $E_A^{(2)} \neq 0$ in optimal ERM models. *Therefore, optimal ERM are not robust against interventional distribution shift.* Also, note that a robust model is not a minimizer of prediction loss on the training distribution as the minimizer in Eq. (4) leads to non-zero $\boldsymbol{\theta}_B^{(A)}$. This can explain the drop in observational accuracy of ERM-Resampled as it improved the interventional accuracy in predicting $A$ in Sec. 3.1. The detailed derivation is provided in App. B.2.

**Minimizing linear dependence:** In Sec. 3.3, we showed that dependence between interventional features correlated positively with the drop in accuracy on interventional data. We will now verify if minimizing dependence between interventional features can minimize the drop in accuracy. The interventional features are given by $\boldsymbol{F}_A = \boldsymbol{\Theta}^{(A)\top} \boldsymbol{X}$ and $\boldsymbol{F}_B' = \boldsymbol{\Theta}^{(B)\top} \boldsymbol{X}$.

$$\boldsymbol{F}_A = \boldsymbol{\Theta}^{(A)\top} \boldsymbol{X} = X_A \boldsymbol{\theta}_A^{(A)} + X_B \boldsymbol{\theta}_B^{(A)}$$
$$\boldsymbol{F}_B' = \boldsymbol{\Theta}^{(B)\top} \boldsymbol{X} = X_A \boldsymbol{\theta}_A^{(B)} + X_B \boldsymbol{\theta}_B^{(B)}$$

For ease of exposition, we will minimize the linear dependence between interventional features instead of enforcing the full statistical independence we described in Sec. 3.2. Following the definition of HSIC (Gretton et al., 2005), the linear dependence in interventional features can be defined as follows[5],

$$\operatorname{Dep}\left(\boldsymbol{F}_A, \boldsymbol{F}_B'\right) = \left\| \mathbb{E}_{P_{\text{int}}} \left[ \boldsymbol{F}_A \boldsymbol{F}_B'^\top \right] \right\|_F^2 \tag{5}$$

Leveraging the independence relations during interventions, we can expand Eq. (5) as,

$$\left\| \mathbb{E}_{P_{\text{int}}} \left[ \boldsymbol{F}_A \boldsymbol{F}_B'^\top \right] \right\|_F^2 = \left\| (w_A^2 \rho_A^2 + \rho_{U_A}^2) \boldsymbol{\theta}_A^{(A)} \boldsymbol{\theta}_A^{(B)\top} + (w_B^2 \rho_{B'}^2 + \rho_{U_B}^2) \boldsymbol{\theta}_B^{(A)} \boldsymbol{\theta}_B^{(B)\top} \right\|_F^2 \tag{6}$$

The dependence loss is thus the Frobenius norm of a sum of rank-one matrices. There are three classes of solutions that minimize Eq. (6): (1) $\boldsymbol{\theta}_A^{(A)} = \boldsymbol{\theta}_B^{(A)} = \boldsymbol{\theta}_A^{(B)} = \boldsymbol{\theta}_B^{(B)} = \boldsymbol{0}$, (2) $\boldsymbol{\theta}_A^{(A)} = \pm \gamma \boldsymbol{\theta}_B^{(A)}$ and $\gamma \boldsymbol{\theta}_A^{(B)} = \mp \boldsymbol{\theta}_B^{(B)}$ for some scalar $\gamma \neq 0$, and (3) $\boldsymbol{\theta}_A^{(A)} = \boldsymbol{0}$ or $\boldsymbol{\theta}_A^{(B)} = \boldsymbol{0}$, and $\boldsymbol{\theta}_B^{(A)} = \boldsymbol{0}$ or $\boldsymbol{\theta}_B^{(B)} = \boldsymbol{0}$. However, all except two of these solutions produce trivial features and increase the classification error. The only remaining non-degenerate solutions are: (S1) $\boldsymbol{\theta}_A^{(A)} = \boldsymbol{0}, \boldsymbol{\theta}_B^{(B)} = \boldsymbol{0}$, and (S2) $\boldsymbol{\theta}_B^{(A)} = \boldsymbol{0}, \boldsymbol{\theta}_A^{(B)} = \boldsymbol{0}$. Note that (S2) corresponds to a robust model. Since both (S1) and (S2) minimize Eq. (5), the solution that minimizes the prediction error on both $A$ and $B$ during training will prevail.

---

[5]For a complete definition of the dependence, refer to App. B.3.

**Proposition 1.** *The total training error for (S1) is strictly greater than that of (S2) when the following conditions are satisfied: (1) $\beta \geq 1 - \frac{1}{|w_{AB}|}$, (2) $\beta \geq \min\left(\frac{\rho_A^2}{2\rho_{B'}^2 + \rho_A^2}, \frac{\rho_{U_A}^2}{w_A^2 w_{AB}^2 \rho_A^2}\right)$.*

Proposition 1 states that a robust model is guaranteed when a minimum amount of interventional data is available during training. Note that Proposition 1 describes sufficient conditions for (S1) to have a larger training error than (S2). In practice, $\beta$ could be smaller. Refer to App. B.3 for a detailed derivation and experimental verification of Proposition 1.

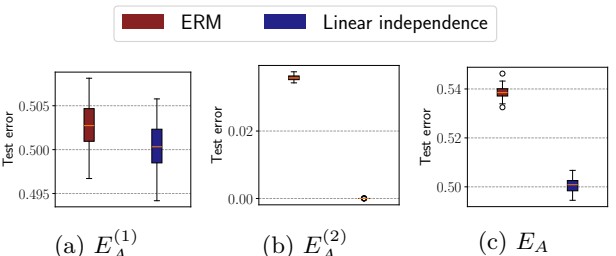

(a) $E_A^{(1)}$     (b) $E_A^{(2)}$     (c) $E_A$

Figure 4: Robust models achieve $E_A^{(2)} = 0$ in Eq. (1). ERM models have a non-zero $\boldsymbol{\theta}_B^{(A)}$ resulting in $E_A^{(2)} \neq 0$. Minimizing linear independence on interventional features results in orthogonal interventional feature spaces where $\boldsymbol{\theta}_B^{(A)} = \boldsymbol{\theta}_A^{(B)} = \mathbf{0}$. Thus, they result in robust models with $E_A^{(2)} = 0$.

**Experimental verification:** To experimentally verify the theoretical results, we simulate the causal model by setting $w_A = w_B = w_{AB} = 1$. The random variables $A$, $B$, $U_A$, and $U_B$ are sampled from independent normal distributions with zero mean and unit variance. We generate $N = 50000$ data points for training with $\beta = 0.5$. The classifiers use 2-dimensional features learned by linear feature extractors to predict $A$ and $B$. The experiment is repeated with 50 seeds. In Eq. (1), the statistical risk was shown to be composed of $E_A^{(1)}$ and $E_A^{(2)}$, plotted in Figs. 4a and 4b respectively. An ideal robust model will achieve $E_A^{(2)} = 0$. As expected, both models have similar $E_A^{(1)}$. However, linear independence models minimize $E_A^{(2)}$, resulting in a lower total error $E_A$ shown in Fig. 4c.

## 4 RepLIn: Enforcing Statistical Independence between Interventional Features

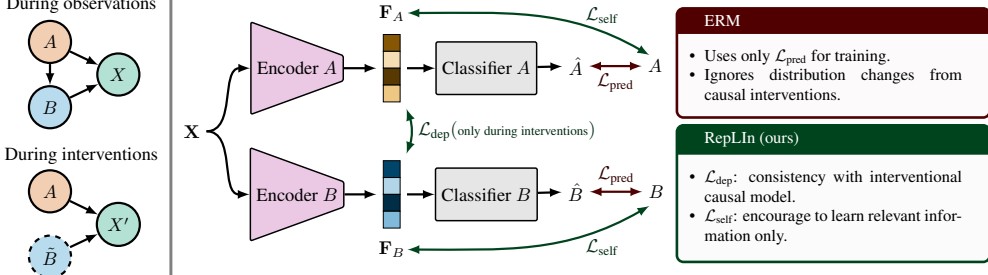

Figure 5: Schematic illustration of **RepLIn** for a causal graph with two attributes ($A \rightarrow B$) and $\boldsymbol{X} = f(A, B, \boldsymbol{U})$. Encoders learn representations $\boldsymbol{F}_A$ and $\boldsymbol{F}_B$ corresponding to $A$ and $B$, which are then used by their corresponding classifiers to predict $\hat{A}$ and $\hat{B}$ respectively. On interventional samples, we minimize $\mathcal{L}_{\text{dep}}$ between the features to ensure their independence. On all samples, we minimize $\mathcal{L}_{\text{self}}$ to encourage the representations to learn only the relevant information.

As noted in the previous section, there is a strong correlation between the drop in accuracy during interventions and interventional feature dependence. We also showed theoretically that minimizing linear dependence between interventional features can improve test time error on interventional data for linear models. Based on this observation, we propose "Representation Learning from Interventional data" (RepLIn) to learn discriminative representations that are robust against interventional distribution shifts.

To enforce independence between interventional features, we propose to use dependence-guided regularization denoted as $\mathcal{L}_{\text{dep}}$ over the prediction loss function (e.g., cross-entropy for classification tasks) used in ERM. We refer to this regularization as "dependence loss" and is defined for the general case in Sec. 3 as

$\mathcal{L}_{\text{dep}} = \sum_{i=1}^{n} \text{NHSIC}(\boldsymbol{F}_{A_i}^{\text{int}}, \boldsymbol{F}_B^{\text{int}})$ . We minimize the dependence loss *only* for the interventional samples in our training set since congruent statistical independence occurs in the data space only during interventions.

However, $\mathcal{L}_{\text{dep}}$ alone is insufficient since learning irrelevant features can minimize $\mathcal{L}_{\text{dep}}$. To avoid such pathological scenarios and encourage the model to learn only relevant information, we introduce another loss that maximizes the dependence between a feature and its corresponding label. We employ this "self-dependence loss" on both observational and interventional data and define it as $\mathcal{L}_{\text{self}} = 1 - \frac{\text{NHSIC}(\boldsymbol{F}_B, B) + \sum_{i=1}^{n} \text{NHSIC}(\boldsymbol{F}_{A_i}, A_i)}{2(n+1)}$ . However, in contrast to $\mathcal{L}_{\text{dep}}$, we use linear kernels in $\mathcal{L}_{\text{self}}$ to maximize a lower estimate of the dependence between the representations and the labels. Using linear kernels in HSIC amounts to $k_P\left(\boldsymbol{x}^{(i)}, \boldsymbol{x}^{(j)}\right) = \boldsymbol{x}^{(i)\top} \boldsymbol{x}^{(j)}$ in Sec. 3.2. In summary, RepLIn optimizes the following total loss: $\mathcal{L}_{\text{total}} = \mathcal{L}_{\text{pred}} + \lambda_{\text{dep}} \mathcal{L}_{\text{dep}} + \lambda_{\text{self}} \mathcal{L}_{\text{self}}$ , where $\lambda_{\text{dep}}$ and $\lambda_{\text{self}}$ are weights that control the contribution of the respective losses. A pictorial overview of the RepLIn pipeline is shown in Fig. 5.

## 5 Experimental Evaluation

In this section, we compare the performance of RepLIn to the baselines on synthetic and real face image datasets. We use the WINDMILL dataset introduced in Sec. 3.1 to verify the effectiveness of RepLIn and evaluate its broader applicability to practical scenarios through the facial attribute prediction task on the CelebA dataset. Our experiments are designed to validate the following hypothesis: *Does explicitly minimizing the interventional feature dependence improve interventional accuracy?*

**Training Hyperparameters and Baselines:** We consider vanilla ERM and ERM-Resampled (Chawla et al., 2002; Cateni et al., 2014) as our primary baselines since they are the most commonly used training algorithms. ERM-Resampled is a strong baseline for group-imbalanced training and domain generalization (Idrissi et al., 2022; Gulrajani & Lopez-Paz, 2021). On WINDMILL dataset, we also consider the following SOTA algorithms in domain generalization: IRMv1 (Arjovsky et al., 2019), Fish (Shi et al., 2022), GroupDRO (Sagawa et al., 2020), SAGM (Wang et al., 2023), DiWA (Rame et al., 2022), and TEP (Qiao & Peng, 2024). The latter two are weight-averaging methods, for which we train 20 independent models per seed. We study two variants of our method: RepLIn and RepLIn-Resampled. The latter variant uses the resampling strategy from ERM-Resampled. In each method, attribute-specific representations are extracted from the input data, which feed into the classifiers to get the final prediction. All baselines share the same architecture for feature extractors. Linear classifiers are used for all baselines. We note that the values of $\lambda_{\text{dep}}$ and $\lambda_{\text{self}}$ in RepLIn variants are kept fixed across all proportions of interventional data $\beta$. A detailed description of the datasets and the training settings is provided in App. A.

**Evaluation Criterion:** Our primary interest is in investigating the accuracy drop when predicting the variables that are unaffected by interventions. Ideally, if the learned features respect causal relations during interventions, we expect no change in the prediction accuracy of parent variables of the intervened variable between observational and interventional distributions. To measure the change, we use the relative drop in accuracy defined in Sec. 3.3: $\text{Rel.}\Delta = \frac{\text{Obs acc.} - \text{Int acc.}}{\text{Obs acc.}}$. Since we optimize NHSIC during training, we use NKCC from Sec. 3.3 to evaluate the dependence between the features on interventional data during testing. We repeat each experiment with five different random seeds and report the mean and standard deviation.

### 5.1 Windmill dataset

We first evaluate our method on the synthetic dataset that helped us identify the relation between the performance gap in predicting $A$ on observational and interventional data in Sec. 3.1. As a reminder, the causal graph consists of two binary random variables $A$ and $B$, where $A \rightarrow B$ during observations. We intervene by setting $B \sim \text{Bernoulli}(0.5)$, breaking the dependence between $A$ and $B$. The proportion of interventional samples in the training data varies from $\beta = 0.01$ to $\beta = 0.5$.

Tab. 2 compares the interventional accuracy of $A$ for various amounts of interventional data. We make the following observations: **(1)** our model outperforms every baseline in interventional accuracy for all values of $\beta$. This clearly demonstrates the advantage of exploiting the underlying causal relations when learning

Accuracy on interventional data. The relative drop in accuracy is shown in parentheses.

| Method | $\beta = 0.5$ | | $\beta = 0.3$ | | $\beta = 0.1$ | | $\beta = 0.05$ | | $\beta = 0.01$ | |
|---|---|---|---|---|---|---|---|---|---|---|
| ERM | 76.87 ± 1.08 | (0.18 ± 0.01) | 69.86 ± 3.19 | (0.29 ± 0.04) | 62.78 ± 1.77 | (0.37 ± 0.02) | 59.52 ± 1.30 | (0.40 ± 0.01) | 60.15 ± 3.12 | (0.40 ± 0.03) |
| ERM-Res. | 73.70 ± 3.19 | (0.22 ± 0.04) | 71.19 ± 3.23 | (0.24 ± 0.03) | 73.62 ± 1.54 | (0.22 ± 0.02) | 71.03 ± 2.83 | (0.25 ± 0.03) | 70.20 ± 3.73 | (0.26 ± 0.03) |
| IRMv1 | 78.24 ± 0.79 | (0.16 ± 0.01) | 74.83 ± 1.74 | (0.20 ± 0.02) | 78.61 ± 2.24 | (0.16 ± 0.02) | 76.28 ± 1.87 | (0.18 ± 0.02) | 71.75 ± 2.03 | (0.24 ± 0.02) |
| Fish | 77.23 ± 2.24 | (0.19 ± 0.02) | 77.23 ± 1.32 | (0.19 ± 0.01) | 78.24 ± 2.09 | (0.18 ± 0.02) | 76.42 ± 1.95 | (0.20 ± 0.02) | 73.92 ± 2.53 | (0.23 ± 0.02) |
| GroupDRO | 80.10 ± 1.66 | (0.02 ± 0.01) | 80.96 ± 1.33 | (0.04 ± 0.02) | 80.35 ± 1.01 | (0.06 ± 0.02) | 77.40 ± 1.16 | (0.08 ± 0.01) | 71.86 ± 1.60 | (0.22 ± 0.02) |
| SAGM | 76.43 ± 2.37 | (0.19 ± 0.02) | 79.05 ± 2.23 | (0.17 ± 0.02) | 76.96 ± 4.36 | (0.18 ± 0.03) | 79.86 ± 1.81 | (0.16 ± 0.02) | 72.81 ± 3.10 | (0.23 ± 0.03) |
| DiWA | 76.61 ± 2.15 | (0.19 ± 0.02) | 76.71 ± 0.59 | (0.19 ± 0.01) | 76.09 ± 0.69 | (0.20 ± 0.01) | 75.83 ± 1.83 | (0.20 ± 0.02) | 73.39 ± 1.31 | (0.22 ± 0.01) |
| TEP | 58.68 ± 4.72 | (0.06 ± 0.19) | 60.42 ± 1.30 | (0.09 ± 0.06) | 56.07 ± 3.35 | (−0.04 ± 0.42) | 58.52 ± 4.36 | (0.01 ± 0.25) | 59.23 ± 1.13 | (0.18 ± 0.11) |
| RepLIn | 87.94 ± 1.46 | (0.08 ± 0.02) | 87.76 ± 2.30 | (0.10 ± 0.02) | 83.23 ± 2.67 | (0.16 ± 0.03) | 73.63 ± 2.43 | (0.25 ± 0.02) | 67.52 ± 2.30 | (0.32 ± 0.03) |
| RepLIn-Res. | 88.46 ± 0.96 | (0.07 ± 0.01) | 88.05 ± 1.04 | (0.08 ± 0.01) | 87.91 ± 1.36 | (0.08 ± 0.01) | 86.38 ± 0.85 | (0.10 ± 0.01) | 78.41 ± 1.27 | (0.18 ± 0.02) |

Table 2: **Results on Windmill dataset:** We evaluate the variants of RepLIn (highlighted in gray) against the baselines on two metrics: interventional accuracy and relative accuracy drop on interventional data compared to observational. As the proportion of interventional data during training ($\beta$) decreases, the problem becomes more challenging. Compared to the baselines, RepLIn maintains its interventional accuracy. A similar trend is observed in the relative accuracy drop, where RepLIn significantly outperforms most baselines. The **best** and the **second-best** results are shown in different colors. "Res." stands for "Resampled".

from interventional data, instead of treating it as a separate domain, and **(2)** comparing ERM and RepLIn with their resampling variants, we observe that resampling is a generally useful technique with large gains when $\beta$ is very small (for example, consider results with $\beta \leq 0.05$). We are also interested in the relative drop in accuracy between observational and interventional data (Rel.$\Delta$). From Tab. 2, we observe that GroupDRO has the lowest Rel.$\Delta$ among the considered methods for $\beta \geq 0.05$, and achieves its best results when more interventional data is available during training. However, this improvement comes at the cost of lower interventional accuracy. Meanwhile, the relative drop in accuracy of RepLIn is comparable to GroupDRO at larger values of $\beta$ and has the least relative drop in accuracy at lower values of $\beta$. DiWA and TEP were provided with the same pool of models trained with minor variations in their hyperparameters. We ignore the Rel.$\Delta$ of TEP since it has very low accuracy, performing barely above random chance. We discuss in Sec. 6.1 how the representations learned by RepLIn are less affected by interventional shifts. As mentioned in Sec. 3.1, interventional robustness may be at odds with observational accuracy as removing spurious information from representations may hurt performance on observational data. We provide the results on observational data in App. D.

## 5.2 Facial Attribute Prediction

We verify the utility of RepLIn for predicting facial attributes on the CelebA dataset (Liu et al., 2015). Images in the CelebA dataset are annotated with 40 labeled binary attributes. We consider two of these attributes – `smiling` and `gender` – as random variables affecting each other causally. Since the true underlying relation between smile and gender is unknown, we adopt the resampling procedure from (Wang & Boddeti, 2022) to induce a desired causal relation between the attributes (`smiling → gender`) and obtain samples. Specifically, to simulate this causal relation, we sample `smiling` from Bernoulli(0.6) first and then sample `gender` according to a probability distribution conditioned on the sampled `smiling` variable. We then sample a face image whose attribute labels match the sampled values. We model the diversity in the images due to unobserved noise variables. Note that, unlike in WINDMILL, the noise variables in this experiment *may be* causally related to the attributes that we wish to predict, adding to the challenges in the dataset. The causal model for this experiment and some sample images are shown in Fig. 7.

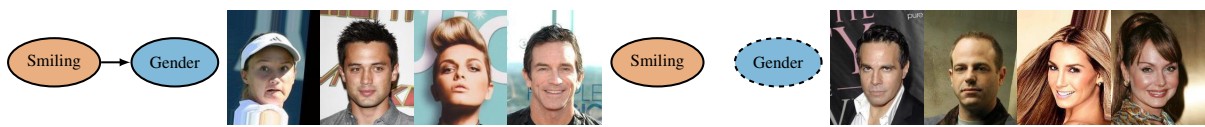

(a) Observational causal graph and samples          (b) Interventional causal graph and samples

Figure 7: Causal model for CelebA before and after intervention along with sample images from these models

Given the face images, we first extract features from ResNet-50 (He et al., 2016) pre-trained on ImageNet dataset (Deng et al., 2009). Then, similar to the architecture for WINDMILL experiments, we employ a shallow MLP to act on these features, followed by a linear classifier to predict the attributes. Our loss functions act upon the features of the MLP. We use 30,000 samples for training and 15,000 for testing. We use the relative drop in interventional accuracy as the primary metric and compare RepLIn-Resampled against ERM-Resampled. We also verify if the correlation between interventional feature dependence and the relative drop in accuracy observed in Sec. 3.3 on WINDMILL experiments holds in a more practical scenario.

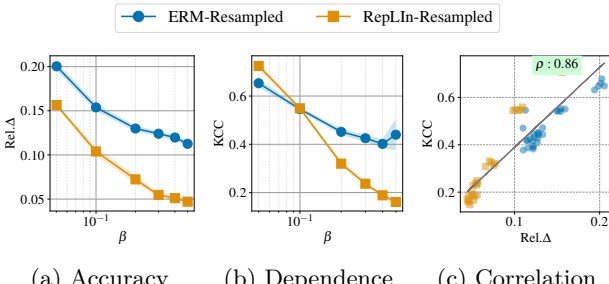

(a) Accuracy     (b) Dependence     (c) Correlation

Figure 8: **Facial Attribute Prediction:** (a) RepLIn has a lower relative drop in accuracy compared to ERM-Resampled. (b) Minimizing interventional feature dependence during training generalizes to testing. (c) Interventional feature dependence correlates positively with the relative drop in accuracy.

Fig. 8 reports the experimental results on facial attribute prediction for various amounts of interventional training data. We make the following observations: **(1)** as the proportion of interventional data increases, the relative drop in accuracy in all methods decreases, **(2)** across all proportions of interventional data, RepLIn consistently outperforms the baseline by $4\% - 7\%$ lower relative drop in accuracy despite the potential challenges due to noise variable being causally related to the attributes of interest, **(3)** relative drop in accuracy and interventional feature dependence show strong positive correlation ($\rho = 0.86$), and **(4)** the interventional feature dependence of RepLIn steadily decreases as the amount of interventional data increases.

## 5.3 Toxicity Prediction in Text

We further evaluate RepLIn on a text classification task on the CivilComments dataset (Borkan et al., 2019). CivilComments consists of comments from online forums, and we use a subset of this dataset labeled with identity attributes (such as "Male", "White", "LGBTQ", etc.) and toxicity scores by humans. The task is to classify each comment as toxic or not. Previous works have identified gender bias in toxicity classifier models (Dixon et al., 2018; Park et al., 2018; Nozza et al., 2019). Therefore, we will simulate a causal model in the training dataset between the attribute "female" and toxicity, similar to Sec. 5.2. During observation, both attributes assume the same binary value. During interventions, toxicity takes value independent of "female". Input text comments are sampled according to these attributes. Similar to our facial attribute prediction experiments, we first extract features from the comments using BERT Devlin et al. (2019) and train the models on these features. Our model architecture consists of a linear layer to learn representations and a linear layer to predict toxicity.

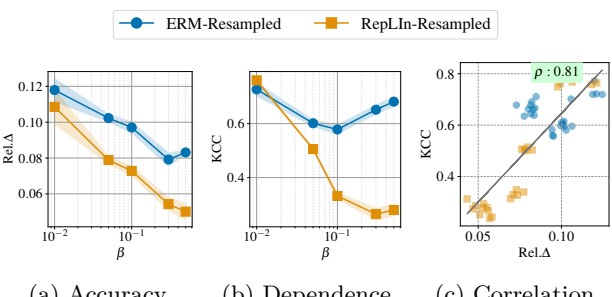

(a) Accuracy     (b) Dependence     (c) Correlation

Figure 9: **Toxicity Prediction in Text:** (a) RepLIn has lower interventional accuracy drop compared to ERM-Resampled; (b) Minimizing $\mathcal{L}_{\text{dep}}$ during training gives us representations that are independent during interventions; (c) The strong correlation between accuracy drop and interventional feature dependence further corroborates our hypothesis in Sec. 3.2.

Fig. 9 compares the performance of RepLIn against ERM-Resampled. Fig. 9b shows that enforcing independence between interventional features minimizes the interventional feature dependence during testing, although its effectiveness drops as $\beta$ approaches 0.01. Yet, RepLIn outperforms the baseline in terms of the accuracy drop during interventions (Fig. 9a).

## 6 Discussion

### 6.1 How does RepLIn improve robustness against interventional distribution shift?

In Sec. 3.4, we showed theoretically that enforcing linear independence between interventional features can improve robustness in a linear model. We verified our claims experimentally in non-linear settings in Sec. 5. In this section, we qualitatively and quantitatively compare the interventional features learned by various methods to understand how RepLIn improves robustness against interventional distribution shift.

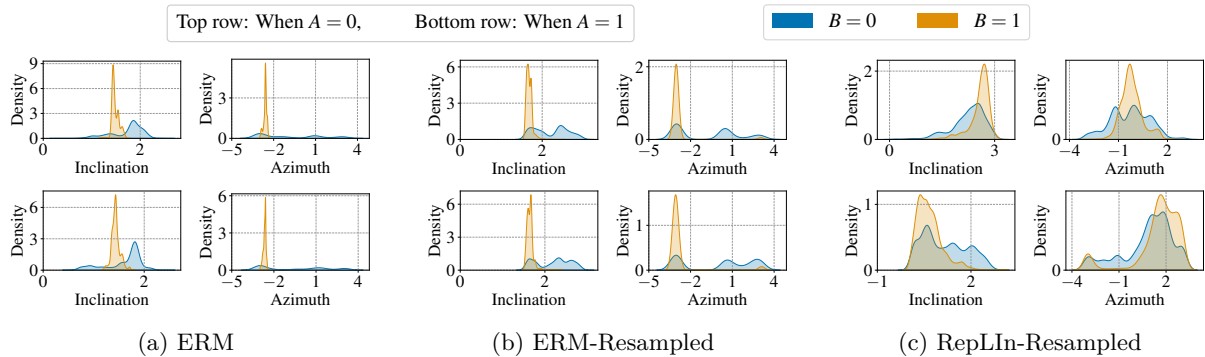

Figure 10: Visualization of interventional features learned by various methods on WINDMILL dataset.

| Method | ERM | ERM-Resampled | IRMv1 | Fish | GroupDRO | RepLIn | RepLIn-Resampled |
|---|---|---|---|---|---|---|---|
| When $A = 0$ | $0.45 \pm 0.058$ | $0.423 \pm 0.105$ | $0.333 \pm 0.122$ | $0.341 \pm 0.111$ | $0.365 \pm 0.066$ | $\mathbf{0.15 \pm 0.03}$ | $\mathbf{0.188 \pm 0.032}$ |
| When $A = 1$ | $0.499 \pm 0.07$ | $0.456 \pm 0.11$ | $0.405 \pm 0.111$ | $0.37 \pm 0.116$ | $0.431 \pm 0.048$ | $\mathbf{0.183 \pm 0.058}$ | $\mathbf{0.168 \pm 0.047}$ |
| Average | $0.475 \pm 0.063$ | $0.439 \pm 0.105$ | $0.369 \pm 0.116$ | $0.355 \pm 0.113$ | $0.398 \pm 0.055$ | $\mathbf{0.166 \pm 0.035}$ | $\mathbf{0.178 \pm 0.036}$ |

Table 3: **Jensen-Shannon (JS) divergence:** The distribution of $\boldsymbol{F}_A^{\text{int}}$ must be invariant to the value assumed by $B$ since $A \perp\!\!\!\perp B$ during interventions. Therefore, JS divergence between $P(\boldsymbol{F}_A^{\text{int}}|B = 0, A = a)$ and $P(\boldsymbol{F}_A^{\text{int}}|B = 1, A = a)$ of a robust model must be zero. We compare the JS divergence between interventional features of the baselines for $\beta = 0.5$. Among the baselines, RepLIn achieves the lowest values of Jensen-Shannon divergence. The lowest and the second lowest scores are highlighted in color.

**Windmill dataset:** We consider the distribution of $\boldsymbol{F}_A^{\text{int}}$ for a fixed value of $A$ and changing values of $B$. Robust representations of $A$ change with $A$ but not $B$. The distribution shift in $\boldsymbol{F}_A^{\text{int}}$ due to changes in $B$ can be quantitatively measured using Jensen-Shannon (JS) divergence. In Tab. 3, we calculate JS divergence between $P(\boldsymbol{F}_A^{\text{int}}|B = 0, A = a)$ and $P(\boldsymbol{F}_A^{\text{int}}|B = 1, A = a)$ for all methods trained on WINDMILL dataset. JS divergence for an ideal robust model must be zero. We observe that $\boldsymbol{F}_A^{\text{int}}$ learned by RepLIn achieves the lowest JS divergence. This shows that $\boldsymbol{F}_A^{\text{int}}$ learned by RepLIn contains the least information about $B$ among the baselines. In our experiments on WINDMILL dataset, all baselines learned 3-dimensional features that lay on a unit radius sphere. Therefore, we can visualize the distributions of their spherical angles, namely inclination and azimuth. We compare the distributions of inclination and azimuth of $\boldsymbol{F}_A^{\text{int}}$ learned by RepLIn-Resampled against the ERM baselines in Fig. 10. Each row shows the distribution of the spherical angles for different values of $A$. Distributions for different values of $B$ have separate colors. These feature distributions for a robust model must change with $A$ but not $B$. We observed from the figure that the feature distributions of the baselines are affected by $B$ and not $A$ due to the dependence between $\boldsymbol{F}_A^{\text{int}}$ and $B$. However, the feature distributions learned by RepLIn change with $A$ and overlap significantly when $B$ takes different values. Thus, our models perform similarly to a robust model. Visualizations of the feature distributions of other baselines are provided in App. E.

**CelebA dataset:** Our learned representations on CelebA are high-dimensional, and therefore we employ Grad-CAM (Selvaraju et al., 2017) to analyze the features and compare them against those learned by ERM-Resampled. Since our primary metric is accuracy in predicting `smiling` during interventions, we visualize

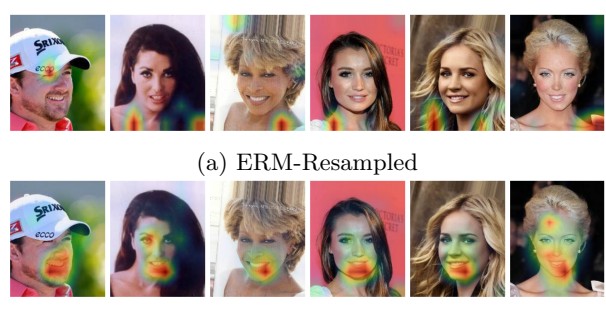

(a) ERM-Resampled

(b) RepLIn-Resampled

Figure 11: Consider these sample face images where the subjects are smiling. The ERM baseline misclassified these samples as not smiling, while RepLIn classified them correctly. We use GradCAM visualizations to identify the regions in the input images that the models used to make their predictions. The ERM model relied on factors such as hair and the presence of a hat that may correlate with gender to predict whether the subjects are smiling. In contrast, RepLIn attended to the lip regions to make predictions.

the parts of the input image that the models attend to for predicting a smile. We consider some samples with smiling $= 1$ that were misclassified by ERM-Resampled but were correctly classified by RepLIn-Resampled. Comparing these predictions would help us explain the robustness of RepLIn. Fig. 11 shows the attention maps from models trained on datasets with 50% interventional data. A robust model would attend to facial regions surrounding lips to make predictions about smiling. Observe that RepLIn-Resampled tends to focus more on the region around the lips while ERM-Resampled attends to other regions of the face.

## 6.2 Scalability with number of nodes

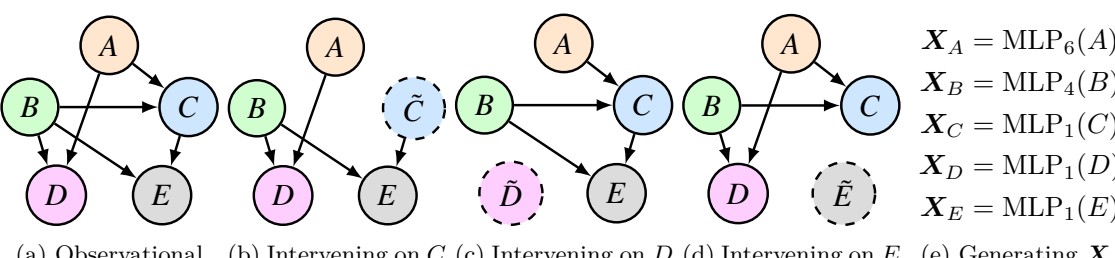

(a) Observational  (b) Intervening on $C$  (c) Intervening on $D$  (d) Intervening on $E$  (e) Generating $\boldsymbol{X}$

$$\boldsymbol{X}_A = \mathrm{MLP}_6(A)$$
$$\boldsymbol{X}_B = \mathrm{MLP}_4(B)$$
$$\boldsymbol{X}_C = \mathrm{MLP}_1(C)$$
$$\boldsymbol{X}_D = \mathrm{MLP}_1(D)$$
$$\boldsymbol{X}_E = \mathrm{MLP}_1(E)$$

Figure 12: **5-variable causal graph**: We construct a 5-variable causal graph to demonstrate the scalability of our method with the number of nodes. To collect interventional data, we intervene on $C$, $D$, and $E$ separately and measure the performance drop in predicting $A$ and $B$ during these interventions. Nodes in the graphs with dashed borders indicate intervened nodes. Note that we do not intervene on multiple targets at a time. The input data signal $\boldsymbol{X}$ is constructed as a concatenation of individual input signals, each being a function of a latent variable, i.e., $\boldsymbol{X} = \begin{bmatrix} \boldsymbol{X}_A^\top & \boldsymbol{X}_B^\top & \boldsymbol{X}_C^\top & \boldsymbol{X}_D^\top & \boldsymbol{X}_E^\top \end{bmatrix}^\top$ Here, $\mathrm{MLP}_l$ indicates a randomly initialized MLP with $l$ linear layers, each followed by a ReLU. We also add Gaussian noise sampled from $\mathcal{N}(0, 0.01)$ to the output of the MLP.

Consider a causal graph with $N$ nodes, each with $K$ parent nodes. To learn robustness against distribution shifts due to interventions on $M$ nodes, RepLIn requires only the independence relations between these $M$ nodes and their parents and the data resulting from separate interventions on these nodes. Therefore, RepLIn can scale with $M$ and $K$. To verify this scalability, we use the causal graph shown in Fig. 12a with five latent variables. It consists of two binary source nodes $A$ and $B$, and three binary derived nodes $C$, $D$, and $E$. During observations, $A$ and $B$ are sampled from independent Bernoulli(0.5) distributions. During observation, the remaining nodes take the following logical expressions: $C := A$ or $B$, $D := A$ and $B$, and $E := \mathtt{not}\ B$ and $C$. Like our previous experiments, the training dataset has interventional data samples collected by intervening on nodes $C$, $D$, and $E$ separately in addition to the observational data. The changes in the causal graph due to these interventions are shown in Figs. 12b to 12d. Each intervened variable assumes values from a Bernoulli(0.5) distribution independent of their parents. Each latent variable $*$ is passed through a randomly initialized MLP with noise added to its output to get a corresponding observed signal $\boldsymbol{X}_*$. These individual signals are concatenated to obtain the observed input signal $\boldsymbol{X}$, as shown in Fig. 12e. The task is to predict the latent variables from the input signal $\boldsymbol{X}$. Since we are interested in the

robustness of the model against interventional distribution shift, our primary metrics will be the predictive accuracy for $A$ and $B$ during interventions on $C$, $D$, and $E$.

Each batch comprises only observational or interventional data after intervention on a single target. Therefore, our method only enforces the independence relations from at most one interventional target in each batch. The validation and test sets consist of samples collected during interventions on $C$, $D$, or $E$. The predictive performances on the test sets are reported in Table 4. We observe that RepLIn significantly improves over the baseline with sufficient interventional data, $\beta > 0.1$. When the proportion of interventional data $\beta \leq 0.1$, RepLIn is comparable with the baseline, suggesting that the benefits of enforcing independence between interventional features extend to larger causal graphs with multiple intervention targets.

| Interventional target | Method | Predictive accuracy on $A$ | | | | Predictive accuracy on $B$ | | | |
|---|---|---|---|---|---|---|---|---|---|
| | | $\beta = 0.5$ | $\beta = 0.3$ | $\beta = 0.1$ | $\beta = 0.05$ | $\beta = 0.5$ | $\beta = 0.3$ | $\beta = 0.1$ | $\beta = 0.05$ |
| $C$ | ERM-Resampled | $79.71 \pm 0.30$ | $76.22 \pm 0.42$ | $\mathbf{73.97 \pm 0.39}$ | $73.56 \pm 0.36$ | $87.60 \pm 0.06$ | $85.45 \pm 0.23$ | $\mathbf{83.89 \pm 0.33}$ | $\mathbf{83.71 \pm 0.40}$ |
| | RepLIn-Resampled | $\mathbf{95.37 \pm 0.97}$ | $\mathbf{78.77 \pm 0.54}$ | $72.15 \pm 0.31$ | $\mathbf{73.74 \pm 0.36}$ | $\mathbf{96.72 \pm 0.81}$ | $\mathbf{86.16 \pm 0.63}$ | $82.35 \pm 0.95$ | $82.43 \pm 0.65$ |
| $D$ | ERM-Resampled | $79.65 \pm 0.43$ | $75.47 \pm 0.64$ | $\mathbf{71.76 \pm 0.35}$ | $\mathbf{70.27 \pm 0.34}$ | $91.05 \pm 0.29$ | $90.21 \pm 0.27$ | $90.36 \pm 0.58$ | $90.55 \pm 0.74$ |
| | RepLIn-Resampled | $\mathbf{95.49 \pm 1.01}$ | $\mathbf{77.76 \pm 0.82}$ | $71.20 \pm 0.82$ | $68.80 \pm 0.79$ | $\mathbf{97.87 \pm 0.31}$ | $\mathbf{92.21 \pm 0.48}$ | $\mathbf{91.40 \pm 0.79}$ | $\mathbf{90.88 \pm 0.89}$ |
| $E$ | ERM-Resampled | $86.63 \pm 0.33$ | $81.90 \pm 0.26$ | $\mathbf{76.20 \pm 0.84}$ | $\mathbf{73.46 \pm 0.37}$ | $81.12 \pm 0.22$ | $78.00 \pm 0.48$ | $\mathbf{74.02 \pm 0.38}$ | $\mathbf{72.97 \pm 0.38}$ |
| | RepLIn-Resampled | $\mathbf{96.71 \pm 0.49}$ | $\mathbf{84.68 \pm 0.36}$ | $75.01 \pm 0.53$ | $71.52 \pm 0.87$ | $\mathbf{96.89 \pm 0.68}$ | $\mathbf{80.88 \pm 0.57}$ | $72.81 \pm 1.13$ | $71.60 \pm 0.59$ |

Table 4: **Results on 5-variable causal graph:** We compare the accuracy of RepLIn in predicting the source nodes $A$ and $B$ during interventions on non-source nodes $C$, $D$, and $E$ against that of ERM-Resampled. Our approach outperforms the baselines with sufficient interventional data.

## 7 Conclusion

This paper considered the problem of learning representations that are robust against interventional distribution shifts by leveraging the statistical independence induced by interventions in the underlying data-generating process. First, we established a strong correlation between the drop in accuracy during interventions and statistical dependence between representations on interventional data. We then showed theoretically that minimizing linear dependence between interventional representations can improve the robustness of a linear model against interventional distribution shift. Building on this result, we proposed RepLIn to learn representations that are robust against interventional distribution shift by explicitly enforcing statistical independence between learned representations on interventional data. Experimental evaluation of RepLIn across different scenarios corresponding to different causal graphs showed that RepLIn can improve predictive accuracy during interventions for various proportions of interventional data. RepLIn is also scalable to the number of causal attributes and can be used with continuous and discrete latent variables. We used qualitative and quantitative tools to show that RepLIn is more successful in learning interventional representations that do not contain information about their child nodes during interventions.

**Limitations:** We assume access to interventional data and information about the intervened node and its parent variables. However, this may be challenging to obtain in practice, especially in safety-critical applications such as drug testing and autonomous driving. In such cases, generative models could generate synthetic interventional data. Another assumption is that hard interventions may not be possible due to real-world experimental constraints. Specialized solutions may be required for such cases. For example, (Bagi et al., 2024) uses a "switch variable" to model soft interventions.

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

## Overview of Appendix

## A  Implementation details

We implement our models using PyTorch (Paszke et al., 2019) and use Adam (Kingma & Ba, 2015) as our optimizer with its default settings. Training hyperparameters for each dataset (such as the number of data points, training epochs, etc.) are shown in Tab. 5. For training stability, we warm up $\lambda_{\mathrm{dep}}$ from 0 to its set value between $sN$ and $eN$ epochs where $N$ is the total number of epochs, and $s$ and $e$ are fractions shown in Tab. 5.

Table 5: List of hyperparameters used for each dataset.

| Dataset | #Training samples | Epochs | Batchsize | Initial LR | Scheduler | $\lambda_{\mathrm{dep}}$ | $\lambda_{\mathrm{self}}$ | Start ($s$) | End ($e$) |
|---|---|---|---|---|---|---|---|---|---|
| WINDMILL | 40,000 | 5000 | 1000 | 2e-3 | MultiStepLR(milestones=[1000], gamma=0.5) | 1 | 1 | 0.66 | 0.99 |
| CelebA | 30,000 | 2000 | 1000 | 1e-3 | MultiStepLR(milestones=[1000], gamma=0.1) | 20 | 2 | 0.01 | 0.99 |

For all methods, we first extract label-specific features from the inputs and pass them through a corresponding classifier to predict the label. The architecture of the feature extractor is the same for all methods on a given dataset, except on the WINDMILL dataset. The classification layer is a linear layer mapping from feature dimensions to the number of classes. The specific details for each dataset are provided below.

**Windmill dataset:** For ERM baselines, we use an MLP with two layers of size 40 and 1, with a ReLU activation after each layer (except the last) to extract the features. However, we observed that enforcing independence using 1-dimensional features was difficult. Therefore, we used 2-dimensional features for RepLIn, which were then normalized to lie on a sphere.

**CelebA dataset:** We first extract features from the raw image using a ResNet-50 (He et al., 2016) pre-trained on ImageNet (Deng et al., 2009). Although these features are not optimal for face attribute prediction, they are useful for face verification (Sharif Razavian et al., 2014). Additionally, it makes the binary attribute prediction task more challenging. We extract attribute-specific features from this input using a linear layer that maps it to a 500-dimensional space.

## B  Theoretical Motivation for RepLIn

In Sec. 3.4, we theoretically motivated RepLIn. This section explains the motivation with detailed proof.

**Sketch of proof:** First, we estimate the statistical risk in predicting the latent variables from interventional data from representations learned by arbitrary linear feature extractors and classifiers. In this statistical risk, we will identify a term that is the source of performance drop during interventions. We will then show that the optimal ERM models will suffer from this performance drop when trained on a dataset comprising observational and interventional data. Finally, we show that minimizing linear dependence between interventional features can lead to robust linear feature extractors.

**Setup:** We follow the same mathematical notation as the main paper, shown in Tab. 6. The input data $\boldsymbol{X}$ is generated as a function of two latent variables of interest, $A$ and $B$. There are noise variables collectively denoted by $\boldsymbol{U}$ that participate in the data generation but are not of learning interest. Our task is to predict $A$ and $B$ from $\boldsymbol{X}$. $A$ and $B$ are causally related during observation. For ease of exposition, we will consider a simple linear relation $B \coloneqq w_{AB}A$. This causal relation breaks when we intervene on $B$. The intervened

| Entity | Notation | Examples |
|---|---|---|
| Scalar | Regular lowercase characters | $a, \gamma$ |
| Random variable | Regular sans-serif uppercase characters | $A$ |
| Random vector | Bold sans-serif uppercase characters | $\boldsymbol{A}$ |
| Distribution of a random variable $A$ | $P$ with subscript | $P_A$ |

Table 6: Mathematical notation used in the proof.

variable is denoted with an added apostrophe (i.e., $B'$). The data generation process can be written in the form of a structural causal model as follows:

$$
\begin{array}{ll}
A \sim P_A & X_A := w_A A + U_A \\
B' \sim P_{B'} & X_B := w_B B + U_B \\
B := w_{AB} A \text{ (during observations)} & \\
B := B' \text{ (during interventions)} & \boldsymbol{X} = \begin{bmatrix} X_A \\ X_B \end{bmatrix} \\
U_A, U_B \sim P_{\mathsf{U}} &
\end{array}
$$

**Training:** The distribution from which training data is sampled is denoted by $P_{\text{train}}$. The training data consists of both observational and interventional samples, which themselves come from distributions $P_{\text{obs}}$ and $P_{\text{int}}$. We are interested in the scenario where $(1 - \beta)$ proportion of the training data is observational, while the remaining $\beta$ proportion is interventional, where $0 < \beta < 1$. The training distribution can be represented as a mixture of observational and interventional distributions as follows:

$$
P_{\text{train}}(\boldsymbol{X}, A, B) = (1 - \beta) P_{\text{obs}}(\boldsymbol{X}, A, B) + \beta P_{\text{int}}(\boldsymbol{X}, A, B)
$$

Typically, we assume $\beta \ll 1$. We will also assume that $A, B, \boldsymbol{U}$, and $\boldsymbol{X}$ have zero mean, so that we may use linear models without bias terms to extract representations corresponding to the variables of interest and train linear classifiers on these representations. The corresponding classifiers are parameterized by $\boldsymbol{c}^{(A)}$ and $\boldsymbol{c}^{(B)}$. The predictions are made by the classifiers from the learned representations as $\hat{A} = \boldsymbol{c}^{(A)\top} \boldsymbol{\Theta}^{(A)\top} \boldsymbol{X}$ and $\hat{B} = \boldsymbol{c}^{(B)\top} \boldsymbol{\Theta}^{(B)\top} \boldsymbol{X}$. The models are trained by minimizing the mean squared error on the training data, $\mathcal{L}_{\text{MSE}} = \mathbb{E}_{P_{\text{train}}} \left[ \left( \left\| A - \hat{A} \right\|_2^2 + \left\| B - \hat{B} \right\|_2^2 \right) \right]$.

### B.1 Statistical Risk in Predicting Interventional Latent Samples

The model predicts $\hat{A}$ and $\hat{B}$ from $\boldsymbol{X}$ during inference. The statistical squared error in predicting $A$ from interventional samples can be written as,

$$
E_A = \mathbb{E}_{P_{\text{int}}} \left[ \left( A - \hat{A} \right)^2 \right] = \mathbb{E}_{P_{\text{int}}} \left[ \left( A - \boldsymbol{c}^{(A)\top} \boldsymbol{\Theta}^{(A)\top} \boldsymbol{X} \right)^2 \right] \tag{7}
$$

The expectation is taken over the interventional distribution over $\boldsymbol{X}, A, B, \boldsymbol{U}$ denoted by $P_{\text{int}}$. $\boldsymbol{\Theta}^{(A)}$ can be written in terms of constituent parameter vectors as $\boldsymbol{\Theta}^{(A)} = \begin{bmatrix} \boldsymbol{\theta}_A^{(A)\top} \\ \boldsymbol{\theta}_B^{(A)\top} \end{bmatrix}$. The predicted latent $\hat{A}$ can hence be

written in terms of these vectors as,

$$\hat{A} = \boldsymbol{c}^{(A)\top}\boldsymbol{\Theta}^{(A)\top}\boldsymbol{X} = \boldsymbol{c}^{(A)\top}\left(X_A\boldsymbol{\theta}_A^{(A)} + X_{B'}\boldsymbol{\theta}_B^{(A)} + \boldsymbol{\Theta}^{(A)\top}\boldsymbol{U}\right)$$

$$= w_A A \boldsymbol{c}^{(A)\top}\boldsymbol{\theta}_A^{(A)} + w_B B' \boldsymbol{c}^{(A)\top}\boldsymbol{\theta}_B^{(A)} + \boldsymbol{c}^{(A)\top}\boldsymbol{\Theta}^{(A)\top}\boldsymbol{U}$$

$$\therefore \left(A - \boldsymbol{c}^{(A)\top}\boldsymbol{\Theta}^{(A)\top}\boldsymbol{X}\right)^2 = \left(\left(1 - w_A\boldsymbol{c}^{(A)\top}\boldsymbol{\theta}_A^{(A)}\right)A + w_B B' \boldsymbol{c}^{(A)\top}\boldsymbol{\theta}_B^{(A)} + \boldsymbol{c}^{(A)\top}\boldsymbol{\Theta}^{(A)\top}\boldsymbol{U}\right)^2$$

$$= \left(1 - w_A\boldsymbol{c}^{(A)\top}\boldsymbol{\theta}_A^{(A)}\right)^2 A^2 + \left(w_B\boldsymbol{c}^{(A)\top}\boldsymbol{\theta}_B^{(A)}\right)^2 B'^2 + \tilde{U}^2$$

$$+ 2\left(1 - w_A\boldsymbol{c}^{(A)\top}\boldsymbol{\theta}_A^{(A)}\right)\left(w_B\boldsymbol{c}^{(A)\top}\boldsymbol{\theta}_B^{(A)}\right)AB'$$

$$+ 2\left(1 - w_A\boldsymbol{c}^{(A)\top}\boldsymbol{\theta}_A^{(A)}\right)\tilde{U}A + 2\left(w_B\boldsymbol{c}^{(A)\top}\boldsymbol{\theta}_B^{(A)}\right)\tilde{U}B' \tag{8}$$

$$\therefore E_A = \mathbb{E}_{P_{\text{int}}}\left[\left(1 - w_A\boldsymbol{c}^{(A)\top}\boldsymbol{\theta}_A^{(A)}\right)^2 A^2 + \left(w_B\boldsymbol{c}^{(A)\top}\boldsymbol{\theta}_B^{(A)}\right)^2 B'^2 + \tilde{U}^2\right]$$

$$+ 2\mathbb{E}_{P_{\text{int}}}\left[\left(1 - w_A\boldsymbol{c}^{(A)\top}\boldsymbol{\theta}_A^{(A)}\right)\left(w_B\boldsymbol{c}^{(A)\top}\boldsymbol{\theta}_B^{(A)}\right)AB'\right]$$

$$+ 2\mathbb{E}_{P_{\text{int}}}\left[\left(1 - w_A\boldsymbol{c}^{(A)\top}\boldsymbol{\theta}_A^{(A)}\right)\tilde{U}A + 2\left(w_B\boldsymbol{c}^{(A)\top}\boldsymbol{\theta}_B^{(A)}\right)\tilde{U}B'\right]$$

where $\tilde{U} = \boldsymbol{c}^{(A)\top}\boldsymbol{\Theta}^{(A)\top}\boldsymbol{U} = \boldsymbol{c}^{(A)\top}\boldsymbol{\theta}_A^{(A)}U_A + \boldsymbol{c}^{(A)\top}\boldsymbol{\theta}_B^{(A)}U_B$. $\boldsymbol{U}$ denotes exogenous variables that are independent of $A$ and $B$. Due to interventions, we also have $A \perp\!\!\!\perp B$. The expectation of $AB'$ will be zero since they are independent and have zero means marginally. Similarly, the expectation of the products of $\tilde{U}$ with $A$ and $B$ will be zero. Therefore,

$$E_A = \underbrace{\left(1 - w_A\boldsymbol{c}^{(A)\top}\boldsymbol{\theta}_A^{(A)}\right)^2 \rho_A^2 + \left(\boldsymbol{c}^{(A)\top}\boldsymbol{\theta}_A^{(A)}\right)^2 \rho_{\boldsymbol{U}_A}^2}_{E_A^{(1)}} + \underbrace{\left(w_B\boldsymbol{c}^{(A)\top}\boldsymbol{\theta}_B^{(A)}\right)^2 \rho_{B'}^2 + \left(\boldsymbol{c}^{(A)\top}\boldsymbol{\theta}_B^{(A)}\right)^2 \rho_{\boldsymbol{U}_B}^2}_{E_A^{(2)}} \tag{9}$$

where $\rho_A^2 = \mathbb{E}_{P_{\text{int}}}\left[A^2\right]$, $\rho_{B'}^2 = \mathbb{E}_{P_{\text{int}}}\left[B'^2\right]$, $\rho_{\boldsymbol{U}_A}^2 = \mathbb{E}_{P_{\text{int}}}\left[U_A^2\right]$, and $\rho_{\boldsymbol{U}_B}^2 = \mathbb{E}_{P_{\text{int}}}\left[U_B^2\right]$.

**Statistical risk for a robust model:** We are interested in robustness against interventional distribution shifts. The predictions of $A$ by a robust model are unaffected by interventions on its child variable $B$. If $\hat{A}$ must not depend on $B'$, then the corresponding representation $\boldsymbol{F}_A$ must not depend on it either, i.e. $\boldsymbol{\theta}_B^{(A)}$ must be a zero vector. Eq. (9) has two terms: $E_A^{(1)}$ and $E_A^{(2)}$. Therefore, a robust model will have $E_A^{(2)} = 0$ since $\boldsymbol{\theta}_B^{(A)} = \boldsymbol{0}$. Therefore, showing that an optimal ERM model has a non-zero $\boldsymbol{\theta}_B^{(A)}$ is sufficient to show that the model is not robust. ~~We will show that an optimal model trained using ERM will have a non-zero $\boldsymbol{\theta}_B^{(A)}$.~~

## B.2  Optimal ERM model

The optimal ERM model can be obtained by minimizing the expected risk in predicting the latent attributes. Since parameters are not shared between the prediction of $\boldsymbol{a}$ and $\boldsymbol{b}$, we can consider their optimization separately. We will consider the optimization of the parameters for predicting $\boldsymbol{a}$ since we are interested in the performance drop in predicting $A$ from interventional data.

$$\boldsymbol{\Theta}^{(A)*}, \boldsymbol{c}^{(A)*} = \underset{\boldsymbol{\Theta}^{(A)}, \boldsymbol{c}^{(A)}}{\arg\min}\; \mathbb{E}_{P_{\text{train}}}\left[\left(A - \boldsymbol{c}^{(A)\top}\boldsymbol{\Theta}^{(A)\top}\boldsymbol{X}\right)^2\right]$$

where $P_{\text{train}}$ is the joint distribution of $(\boldsymbol{X}, A, B)$ during training. As mentioned earlier, $P_{\text{train}}$ is a mixture of observational distribution $P_{\text{obs}}$ and interventional distribution $P_{\text{int}}$, with $(1-\beta)$ and $\beta$ acting as the mixture weights. Therefore, the training objective can be rewritten as,

$$\boldsymbol{\Theta}^{(A)*}, \boldsymbol{c}^{(A)*} = \underset{\boldsymbol{\Theta}^{(A)}, \boldsymbol{c}^{(A)}}{\arg\min}\; J(\boldsymbol{\Theta}^{(A)}, \boldsymbol{c}^{(A)})$$

$$\text{where,}\; J(\boldsymbol{\Theta}^{(A)}, \boldsymbol{c}^{(A)}) = \left((1-\beta)\mathbb{E}_{P_{\text{obs}}}\left[\left(A - \boldsymbol{c}^{(A)\top}\boldsymbol{\Theta}^{(A)\top}\boldsymbol{X}\right)^2\right] + \beta\mathbb{E}_{P_{\text{int}}}\left[\left(A - \boldsymbol{c}^{(A)\top}\boldsymbol{\Theta}^{(A)\top}\boldsymbol{X}\right)^2\right]\right) \tag{10}$$

Expanding the error term on observational data, we have,

$$c^{(A)\top}\Theta^{(A)\top}X = c^{(A)\top}\left(X_A\theta_A^{(A)} + X_B\theta_B^{(A)} + \Theta^{(A)\top}U\right)$$

$$= w_A A c^{(A)\top}\theta_A^{(A)} + w_B B c^{(A)\top}\theta_B^{(A)} + c^{(A)\top}\Theta^{(A)\top}U$$

$$= w_A A c^{(A)\top}\theta_A^{(A)} + w_B w_{AB} A c^{(A)\top}\theta_B^{(A)} + c^{(A)\top}\Theta^{(A)\top}U$$

$$\therefore \left(A - c^{(A)\top}\Theta^{(A)\top}X\right)^2 = \left(A - w_A A c^{(A)\top}\theta_A^{(A)} - w_B w_{AB} A c^{(A)\top}\theta_B^{(A)} - c^{(A)\top}\Theta^{(A)\top}U\right)^2$$

$$= \left(\left(1 - w_A c^{(A)\top}\theta_A^{(A)} - w_B w_{AB} c^{(A)\top}\theta_B^{(A)}\right)A - c^{(A)\top}\Theta^{(A)\top}U\right)^2$$

$$= \left(1 - w_A c^{(A)\top}\theta_A^{(A)} - w_B w_{AB} c^{(A)\top}\theta_B^{(A)}\right)^2 A^2 + \tilde{U}^2$$

$$- 2\left(1 - w_A c^{(A)\top}\theta_A^{(A)} - w_B w_{AB} c^{(A)\top}\theta_B^{(A)}\right)A\tilde{U}$$

where $\tilde{U} = c^{(A)\top}\Theta^{(A)\top}U = U_A c^{(A)\top}\theta_A^{(A)} + U_B c^{(A)\top}\theta_B^{(A)}$ from App. B.1. Since the exogenous variable $U$ is independent of $A$ and $B$, the expectation of their products over the observational distribution becomes zero. Therefore,

$$\mathbb{E}_{P_{\text{obs}}}\left[\left(A - c^{(A)\top}\Theta^{(A)\top}X\right)^2\right] = \left(1 - w_A c^{(A)\top}\theta_A^{(A)} - w_B w_{AB} c^{(A)\top}\theta_B^{(A)}\right)^2 \mathbb{E}_{P_{\text{obs}}}\left[A^2\right] + \mathbb{E}_{P_{\text{obs}}}\left[\tilde{U}^2\right]$$

$$= \left(1 - w_A c^{(A)\top}\theta_A^{(A)} - w_B w_{AB} c^{(A)\top}\theta_B^{(A)}\right)^2 \rho_A^2 + \left(c^{(A)\top}\theta_A^{(A)}\right)^2 \rho_{U_A}^2 + \left(c^{(A)\top}\theta_B^{(A)}\right)^2 \rho_{U_B}^2$$

$$(11)$$

Note that, $\rho_A^2 = \mathbb{E}_{P_{\text{obs}}}\left[A^2\right]$, $\rho_{U_A}^2 = \mathbb{E}_{P_{\text{obs}}}\left[U_A^2\right]$, and $\rho_{U_B}^2 = \mathbb{E}_{P_{\text{obs}}}\left[U_B^2\right]$ similar to App. B.1 since these values are unaffected by interventions. The expansion of the error term on interventional data was derived in Eq. (9). Thus, the overall training objective Eq. (10) can be written as,

$$J(\Theta^{(A)}, c^{(A)}) = (1-\beta)\left(\left(1 - w_A c^{(A)\top}\theta_A^{(A)} - w_B w_{AB} c^{(A)\top}\theta_B^{(A)}\right)^2 \rho_A^2 + \left(c^{(A)\top}\theta_A^{(A)}\right)^2 \rho_{U_A}^2 + \left(c^{(A)\top}\theta_B^{(A)}\right)^2 \rho_{U_B}^2\right)$$

$$+ \beta\left(\left(1 - w_A c^{(A)\top}\theta_A^{(A)}\right)^2 \rho_A^2 + \left(w_B c^{(A)\top}\theta_B^{(A)}\right)^2 \rho_{B'}^2 + \left(c^{(A)\top}\theta_A^{(A)}\right)^2 \rho_{U_A}^2 + \left(c^{(A)\top}\theta_B^{(A)}\right)^2 \rho_{U_B}^2\right)$$

We set $\psi_1 = c^{(A)\top}\theta_A^{(A)}$ and $\psi_2 = c^{(A)\top}\theta_B^{(A)}$. Since ERM jointly optimizes the feature extractors and the classifiers, no unique solution minimizes the prediction loss. For example, scaling the feature extractor parameters by an arbitrary constant scalar $\gamma$ and the classifier parameters by $1/\gamma$ will give the same error. Therefore, we can optimize $J(\Theta^{(A)}, c^{(A)})$ over $\psi_1$ and $\psi_2$, similar to (Arjovsky et al., 2019).

$$J(\Theta^{(A)}, c^{(A)}) = (1-\beta)\left((1 - w_A\psi_1 - w_B w_{AB}\psi_2)^2 \rho_A^2 + \psi_1^2\rho_{U_A}^2 + \psi_2^2\rho_{U_B}^2\right)$$

$$+ \beta\left((1 - w_A\psi_1)^2 \rho_A^2 + w_B^2\psi_2^2\rho_{B'}^2 + \psi_1^2\rho_{U_A}^2 + \psi_2^2\rho_{U_B}^2\right)$$

$$(12)$$

The optimal values of $\psi_1$ and $\psi_2$ are the stationary points of $J(\Theta^{(A)}, c^{(A)})$ (denoted by $J$ for brevity). Thus the optimal parameter values can be solved for by taking the first-order derivatives of $J$ w.r.t. $\psi_1$ and $\psi_2$ and setting them to zero.

$$\frac{\partial J}{\partial \psi_1} = 2(1-\beta)\left(-(1 - w_A\psi_1 - w_B w_{AB}\psi_2)w_A\rho_A^2 + \psi_1\rho_{U_A}^2\right) + 2\beta\left(-(1 - w_A\psi_1)w_A\rho_A^2 + \psi_1\rho_{U_A}^2\right)$$

$$\frac{\partial J}{\partial \psi_2} = 2(1-\beta)\left(-(1 - w_A\psi_1 - w_B w_{AB}\psi_2)w_B w_{AB}\rho_A^2 + \psi_2\rho_{U_B}^2\right) + 2\beta\left(w_B^2\psi_2\rho_{B'}^2 + \psi_2\rho_{U_B}^2\right)$$

Setting $\frac{\partial J}{\partial \psi_1} = \frac{\partial J}{\partial \psi_2} = 0$, we have,

$$\begin{aligned}
\left(w_A^2 \rho_A^2 + \rho_{\boldsymbol{U}_A}^2\right)\psi_1 && +(1-\beta)w_A w_B w_{AB}\rho_A^2\psi_2 && -w_A\rho_A^2 &= 0 \\
(1-\beta)w_A w_B w_{AB}\rho_A^2\psi_1 && +\left(\beta w_B^2 \rho_{B'}^2 + (1-\beta)w_B^2 w_{AB}^2\rho_A^2 + \rho_{\boldsymbol{U}_B}^2\right)\psi_2 && -(1-\beta)w_B w_{AB}\rho_A^2 &= 0
\end{aligned}$$

The equations are of the form $u_1\psi_1 + v_1\psi_2 + w_1 = 0$ and $u_2\psi_1 + v_2\psi_2 + w_2 = 0$. We can solve for $\psi_2$ as $\psi_2 = \frac{w_2 u_1 - w_1 u_2}{v_1 u_2 - v_2 u_1}$. Since we are only interested in probing the robustness of ERM models, we will check if $\psi_2$ is zero instead of fully solving the system of linear equations. $E_{\mathsf{A}}^{(2)}$ in Eq. (9) is zero if $\psi_2 = 0$, i.e. if $w_2 u_1 - w_1 u_2 = 0$.

$$\begin{aligned}
w_2 u_1 - w_1 u_2 &= -(1-\beta)w_B w_{AB}\left(w_A^2\rho_A^2 + \rho_{\boldsymbol{U}_A}^2\right)\rho_A^2 + 4(1-\beta)w_A^2 w_B w_{AB}\rho_A^4 \\
&= -(1-\beta)w_B w_{AB}\rho_A^2\rho_{\boldsymbol{U}_A}^2
\end{aligned}$$

Unless the training data is entirely composed of interventional data (i.e., $\beta = 1$), $w_2 u_1 - w_1 u_2$ is not zero. Thus, the optimal ERM model is not robust against interventional distribution shifts.

### B.3 Minimizing Linear Dependence

In Sec. 3.3, we showed that dependence between interventional features correlated positively with the drop in accuracy on interventional data. We will now verify if minimizing dependence between interventional features can minimize the drop in accuracy. For ease of exposition, we will minimize the linear dependence between interventional features instead of enforcing statistical independence. The interventional features are given by $\boldsymbol{F}_A = \boldsymbol{\Theta}^{(A)\top}\boldsymbol{X}$ and $\boldsymbol{F}_B' = \boldsymbol{\Theta}^{(B)\top}\boldsymbol{X}$.

$$\begin{aligned}
\boldsymbol{F}_A = \boldsymbol{\Theta}^{(A)\top}\boldsymbol{X} &= \begin{bmatrix} \boldsymbol{\theta}_A^{(A)} & \boldsymbol{\theta}_B^{(A)} \end{bmatrix}\begin{bmatrix} X_A \\ X_B \end{bmatrix} \\
&= X_A\boldsymbol{\theta}_A^{(A)} + X_B\boldsymbol{\theta}_B^{(A)} \\
\boldsymbol{F}_B' = \boldsymbol{\Theta}^{(B)\top}\boldsymbol{X} &= \begin{bmatrix} \boldsymbol{\theta}_A^{(B)} & \boldsymbol{\theta}_B^{(B)} \end{bmatrix}\begin{bmatrix} X_A \\ X_B \end{bmatrix} \\
&= X_A\boldsymbol{\theta}_A^{(B)} + X_B\boldsymbol{\theta}_B^{(B)}
\end{aligned}$$

To define linear independence between interventional features, we use the following definition of cross-covariance from (Gretton et al., 2005):

**Definition 1.** *The* cross-covariance operator *associated with the joint probability $p_{XY}$ is a linear operator $C_{XY} : \mathcal{G} \to \mathcal{F}$ defined as*

$$C_{XY} = \mathbb{E}_{XY}\left[(\phi(X) - \mu_X) \otimes (\psi(Y) - \mu_Y)\right]$$

*where $\mathcal{G}$ and $\mathcal{F}$ are reproducing kernel Hilbert spaces (RKHSs) defined by feature maps $\phi$ and $\psi$ respectively, and $\otimes$ is the tensor product defined as follows*

$$(f \otimes g)h \coloneqq f\langle g, h\rangle_{\mathcal{G}} \text{ for all } h \in \mathcal{G}$$

*where $\langle \cdot, \cdot \rangle$ is the inner product defined over $\mathcal{G}$.*

In our case, instead of RKHS, we have finite-dimensional feature space $\mathbb{R}^d$. Therefore, we have the cross-covariance matrix as follows,

$$C_{XY} = \mathbb{E}_{XY}\left[\phi(X) \otimes \psi(Y)\right] = \mathbb{E}_{XY}\left[\phi(X)\psi(Y)^\top\right]$$

given that the feature maps have zero mean. Following the definition of HSIC (Gretton et al., 2005), linear dependence in the finite-dimensional case between $X$ and $Y$ is defined as the Frobenius norm of the cross-covariance matrix. Therefore, we define the linear dependence loss between the interventional features as,

$$\mathcal{L}_{\text{dep}} = \text{Dep}\left(\boldsymbol{F}_A, \boldsymbol{F}_B'\right) = \left\|\mathbb{E}_{P_{\text{int}}}\left[\boldsymbol{F}_A\boldsymbol{F}_B'^\top\right]\right\|_F^2 \tag{13}$$

Leveraging the independence relations during interventions, we can expand Eq. (13) as,

$$\mathbb{E}_{P_{\text{int}}}\left[\boldsymbol{F}_A\boldsymbol{F}_B'^\top\right] = \mathbb{E}_{P_{\text{int}}}\left[\left(X_A\boldsymbol{\theta}_A^{(A)} + X_B\boldsymbol{\theta}_B^{(A)}\right)\left(X_A\boldsymbol{\theta}_A^{(B)} + X_B\boldsymbol{\theta}_B^{(B)}\right)^\top\right]$$

$$= \mathbb{E}_{P_{\text{int}}}\left[X_A^2\boldsymbol{\theta}_A^{(A)}\boldsymbol{\theta}_A^{(B)\top} + X_AX_B\boldsymbol{\theta}_A^{(A)}\boldsymbol{\theta}_B^{(B)\top} + X_AX_B\boldsymbol{\theta}_B^{(A)}\boldsymbol{\theta}_A^{(B)\top} + X_B^2\boldsymbol{\theta}_A^{(A)}\boldsymbol{\theta}_B^{(B)\top}\right]$$

$$= (w_A^2\rho_A^2 + \rho_{\boldsymbol{U}_A}^2)\boldsymbol{\theta}_A^{(A)}\boldsymbol{\theta}_A^{(B)\top} + (w_B^2\rho_{B'}^2 + \rho_{\boldsymbol{U}_B}^2)\boldsymbol{\theta}_B^{(A)}\boldsymbol{\theta}_B^{(B)\top}$$

$$\therefore \mathcal{L}_{\text{dep}} = \left\|(w_A^2\rho_A^2 + \rho_{\boldsymbol{U}_A}^2)\boldsymbol{\theta}_A^{(A)}\boldsymbol{\theta}_A^{(B)\top} + (w_B^2\rho_{B'}^2 + \rho_{\boldsymbol{U}_B}^2)\boldsymbol{\theta}_B^{(A)}\boldsymbol{\theta}_B^{(B)\top}\right\|_F^2$$

In the last step, all cross-covariance terms are zero due to the independence of the corresponding random variables in the causal graph. The dependence loss is the Frobenius norm of a sum of rank-one matrices $\boldsymbol{\theta}_A^{(A)}\boldsymbol{\theta}_A^{(B)\top}$ and $\boldsymbol{\theta}_B^{(A)}\boldsymbol{\theta}_B^{(B)\top}$. Consider the following general form: $\boldsymbol{Z} = \boldsymbol{a}\boldsymbol{b}^\top + \boldsymbol{c}\boldsymbol{d}^\top$. Then $Z_{ij} = a_ib_j + c_id_j$.

$$\|\boldsymbol{Z}\|_F^2 = \sum_{ij}(a_ib_j + c_id_j)^2$$

$\|\boldsymbol{Z}\|_F^2$ is a sum of squares and thus is zero iff $a_ib_j + c_id_j = 0$, $\forall i, j$. Therefore, $\mathcal{L}_{\text{dep}}$ is minimized when $\theta_{Ai}^{(A)}\theta_{Aj}^{(B)} + \theta_{Bi}^{(A)}\theta_{Bj}^{(B)} = 0$, $\forall i, j$. The potential solutions that minimize $\mathcal{L}_{\text{dep}}$ are (1) $\boldsymbol{\theta}_A^{(A)} = \boldsymbol{\theta}_B^{(A)} = \boldsymbol{\theta}_A^{(B)} = \boldsymbol{\theta}_B^{(B)} = \boldsymbol{0}$, (2) $\boldsymbol{\theta}_A^{(A)} = \pm\gamma\boldsymbol{\theta}_B^{(A)}$ and $\gamma\boldsymbol{\theta}_A^{(B)} = \mp\boldsymbol{\theta}_B^{(B)}$ for some $\gamma \neq 0$, and (3) $\boldsymbol{\theta}_A^{(A)} = \boldsymbol{0}$ or $\boldsymbol{\theta}_A^{(B)} = \boldsymbol{0}$, and $\boldsymbol{\theta}_B^{(A)} = \boldsymbol{0}$ or $\boldsymbol{\theta}_B^{(B)} = \boldsymbol{0}$. The former two solutions result in trivial features and will increase the classification error. The latter solution contains four possible solutions, out of which two solutions result in trivial features. Solutions resulting in trivial features are unlikely to occur during optimization due to a large classification error. Therefore, we need to consider only the remaining two solutions.

The possible solutions are: (1) $\boldsymbol{\theta}_A^{(A)} = \boldsymbol{0}, \boldsymbol{\theta}_B^{(B)} = \boldsymbol{0}$, and (2) $\boldsymbol{\theta}_B^{(A)} = \boldsymbol{0}, \boldsymbol{\theta}_A^{(B)} = \boldsymbol{0}$. Intuitively, in the former solution, $A$ and $B$ will be predicted using $X_B$ and $X_A$ respectively, and the latter solution corresponds to a robust feature extractor that minimizes the reducible error in Eq. (9). We will compare the predictive error achieved by these solutions to compare their likelihood during training.

Recall the expression for training error in predicting $A$ from Eq. (12).

$$J_A(\boldsymbol{\Theta}^{(A)}, \boldsymbol{c}^{(A)}) = (1-\beta)\left((1 - w_A\psi_{A1} - w_Bw_{AB}\psi_{A2})^2\rho_A^2 + \psi_{A1}^2\rho_{\boldsymbol{U}_A}^2 + \psi_{A2}^2\rho_{\boldsymbol{U}_B}^2\right)$$

$$+ \beta\left((1 - w_A\psi_{A1})^2\rho_A^2 + w_B^2\psi_{A2}^2\rho_{B'}^2 + \psi_{A1}^2\rho_{\boldsymbol{U}_A}^2 + \psi_{A2}^2\rho_{\boldsymbol{U}_B}^2\right)$$

$$= (1-\beta)\left((1 - w_A\psi_{A1} - w_Bw_{AB}\psi_{A2})^2\rho_A^2\right)$$

$$+ \beta\left((1 - w_A\psi_{A1})^2\rho_A^2 + w_B^2\psi_{A2}^2\rho_{B'}^2\right) + \psi_{A1}^2\rho_{\boldsymbol{U}_A}^2 + \psi_{A2}^2\rho_{\boldsymbol{U}_B}^2$$

We use $\psi_{A1}$ and $\psi_{A2}$ instead of $\psi_1$ and $\psi_2$ respectively to denote the parameters for predicting $A$. A similar expression can be written for the error in predicting $B$ with $\psi_{B1}$ and $\psi_{B2}$ denoting the parameters for predicting $B$.

$$J_B(\boldsymbol{\Theta}^{(B)}, \boldsymbol{c}^{(B)}) = (1-\beta)\left((1 - w_A\psi_{B1} - w_Bw_{AB}\psi_{B2})^2\rho_A^2 + \psi_{B1}^2\rho_{\boldsymbol{U}_A}^2 + \psi_{B2}^2\rho_{\boldsymbol{U}_B}^2\right)$$

$$+ \beta\left(w_A^2\psi_{B1}^2\rho_A^2 + (1 - w_B\psi_{B2})^2\rho_{B'}^2 + \psi_{B1}^2\rho_{\boldsymbol{U}_A}^2 + \psi_{B2}^2\rho_{\boldsymbol{U}_B}^2\right)$$

$$= (1-\beta)\left((1 - w_A\psi_{B1} - w_Bw_{AB}\psi_{B2})^2\rho_A^2\right)$$

$$+ \beta\left(w_A^2\psi_{B1}^2\rho_A^2 + (1 - w_B\psi_{B2})^2\rho_{B'}^2\right) + \psi_{B1}^2\rho_{\boldsymbol{U}_A}^2 + \psi_{B2}^2\rho_{\boldsymbol{U}_B}^2$$

**Case 1: When $\boldsymbol{\theta}_A^{(A)} = \boldsymbol{0}, \boldsymbol{\theta}_B^{(B)} = \boldsymbol{0}$:** In this case, $\psi_{A1} = 0$ and $\psi_{B2} = 0$. Therefore, the predictive error during training for each latent variable can be written as,

$$J_A = (1-\beta)(w_Bw_{AB}\psi_{A2} - 1)^2\rho_A^2 + \beta\rho_A^2 + \beta w_B^2\psi_{A2}^2\rho_{B'}^2 + \psi_{A2}^2\rho_{\boldsymbol{U}_B}^2$$

$$J_B = (1-\beta)(w_A\psi_{B1} - w_{AB})^2\rho_A^2 + \beta w_A^2\psi_{B1}^2\rho_A^2 + \beta\rho_{B'}^2 + \psi_{B1}^2\rho_{\boldsymbol{U}_A}^2$$

The optimal values of $\psi_{A2}$ and $\psi_{B1}$ can be obtained by equating the gradients of $R_A$ and $R_B$ to zero.

$$\frac{\partial J_A}{\partial \psi_{A2}} = 2(1-\beta)w_B w_{AB} \left(w_B w_{AB}\psi_{A2} - 1\right)\rho_A^2 + 2\beta w_B^2 \psi_{A2}\rho_{B'}^2 + 2\psi_{A2}\rho_{U_B}^2 = 0$$

$$\therefore \psi_{A2}^* = \frac{(1-\beta)w_B w_{AB}\rho_A^2}{(1-\beta)w_B^2 w_{AB}^2 \rho_A^2 + \beta w_B^2 \rho_{B'}^2 + \rho_{U_B}^2}$$

$$J_A^* = \frac{(1-\beta)\rho_A^2 \left(\beta w_B^2 \rho_{B'}^2 + \rho_{U_B}^2\right)}{(1-\beta)w_B^2 w_{AB}^2 \rho_A^2 + \beta w_B^2 \rho_{B'}^2 + \rho_{U_B}^2} + \beta\rho_A^2$$

$$\frac{\partial J_B}{\partial \psi_{B1}} = 2(1-\beta)w_A \left(w_A\psi_{B1} - w_{AB}\right)\rho_A^2 + 2\beta w_A^2 \psi_{B1}\rho_A^2 + 2\psi_{B1}\rho_{U_A}^2 = 0$$

$$\therefore \psi_{B1}^* = \frac{(1-\beta)w_A w_{AB}\rho_A^2}{w_A^2 \rho_A^2 + \rho_{U_A}^2}$$

$$J_B^* = \frac{(1-\beta)w_{AB}^2 \rho_A^2(\beta w_A^2 \rho_A^2 + \rho_{U_A}^2)}{w_A^2 \rho_A^2 + \rho_{U_A}^2} + \beta\rho_{B'}^2$$

The combined training error for this solution is,

$$J_1^* = J_A^* + J_B^*$$

$$= \frac{(1-\beta)\rho_A^2 \left(\beta w_B^2 \rho_{B'}^2 + \rho_{U_B}^2\right)}{(1-\beta)w_B^2 w_{AB}^2 \rho_A^2 + \beta w_B^2 \rho_{B'}^2 + \rho_{U_B}^2} + \beta\rho_A^2$$

$$+ \frac{(1-\beta)w_{AB}^2 \rho_A^2(\beta w_A^2 \rho_A^2 + \rho_{U_A}^2)}{w_A^2 \rho_A^2 + \rho_{U_A}^2} + \beta\rho_{B'}^2 \tag{14}$$

**Case 2: When $\boldsymbol{\theta}_B^{(A)} = \mathbf{0}, \boldsymbol{\theta}_A^{(B)} = \mathbf{0}$:** Here, $\psi_{A2} = 0$ and $\psi_{B1} = 0$. The predictive error during training for each latent variable can be written as,

$$J_A = (w_A\psi_{A1} - 1)^2 \rho_A^2 + \psi_{A1}^2 \rho_{U_A}^2$$

$$J_B = \left((1-\beta)w_{AB}^2 \rho_A^2 + \beta\rho_{B'}^2\right)(w_B\psi_{B2} - 1)^2 + \psi_{B2}^2 \rho_{U_B}^2$$

We follow the former procedure to estimate the optimal values of $\psi_{A1}$ and $\psi_{B2}$.

$$\frac{\partial J_A}{\partial \psi_{A1}} = 2w_A \left(w_A\psi_{A1} - 1\right)\rho_A^2 + 2\psi_{A1}\rho_{U_A}^2 = 0$$

$$\therefore \psi_{A1}^* = \frac{w_A\rho_A^2}{w_A^2 \rho_A^2 + \rho_{U_A}^2}$$

$$J_A^* = \frac{\rho_A^2 \rho_{U_A}^2}{w_A^2 \rho_A^2 + \rho_{U_A}^2}$$

$$\frac{\partial J_B}{\partial \psi_{B2}} = 2w_B \left((1-\beta)w_{AB}^2 \rho_A^2 + \beta\rho_{B'}^2\right)(w_B\psi_{B2} - 1) + 2\psi_{B2}\rho_{U_B}^2$$

$$\therefore \psi_{B2}^* = \frac{(1-\beta)w_B w_{AB}^2 \rho_A^2 + \beta w_B \rho_{B'}^2}{(1-\beta)w_B^2 w_{AB}^2 \rho_A^2 + \beta w_B^2 \rho_{B'}^2 + \rho_{U_B}^2}$$

$$J_B^* = \frac{\left((1-\beta)w_{AB}^2 \rho_A^2 + \beta\rho_{B'}^2\right)\rho_{U_B}^2}{(1-\beta)w_B^2 w_{AB}^2 \rho_A^2 + \beta w_B^2 \rho_{B'}^2 + \rho_{U_B}^2}$$

The combined training error for this solution is,

$$J_2^* = J_A^* + J_B^*$$

$$= \frac{\rho_A^2 \rho_{U_A}^2}{w_A^2 \rho_A^2 + \rho_{U_A}^2} + \frac{\left((1-\beta)w_{AB}^2 \rho_A^2 + \beta\rho_{B'}^2\right)\rho_{U_B}^2}{(1-\beta)w_B^2 w_{AB}^2 \rho_A^2 + \beta w_B^2 \rho_{B'}^2 + \rho_{U_B}^2} \tag{15}$$

Comparing $J_1^*$ and $J_2^*$,

$$J_1^* - J_2^* = \frac{(1-\beta)\beta w_B^2 \rho_A^2 \rho_{B'}^2 + (1-\beta)\rho_A^2 \rho_{\boldsymbol{U}_B}^2 - (1-\beta)w_{AB}^2 \rho_A^2 \rho_{\boldsymbol{U}_B}^2 - \beta \rho_{B'}^2 \rho_{\boldsymbol{U}_B}^2}{(1-\beta)w_B^2 w_{AB}^2 \rho_A^2 + \beta w_B^2 \rho_{B'}^2 + \rho_{\boldsymbol{U}_B}^2}$$
$$+ \frac{(1-\beta)\beta w_A^2 w_{AB}^2 \rho_A^4 + (1-\beta)w_{AB}^2 \rho_A^2 \rho_{\boldsymbol{U}_A}^2 - \rho_A^2 \rho_{\boldsymbol{U}_A}^2}{w_A^2 \rho_A^2 + \rho_{\boldsymbol{U}_A}^2} + \beta(\rho_A^2 + \rho_{B'}^2)$$

Simplifying the above expression, we get the condition that $J_1^* - J_2^* > 0$ if $\beta$ satisfies the following conditions:
(1) $\beta \geq 1 - \frac{1}{|w_{AB}|}$, (2) $\beta \geq \min\left(\frac{\rho_A^2}{2\rho_{B'}^2 + \rho_A^2}, \frac{\rho_{\boldsymbol{U}_A}^2}{w_A^2 w_{AB}^2 \rho_A^2}\right)$. The conditions imply that enforcing linear independence results in robust feature extractors when *enough* interventional data is available during training.

However, this is only a sufficient condition that strictly ensures $J_1^* - J_2^* > 0$. In practice, $\beta$ could be much lower, especially when the total loss is of the form $\mathcal{L}_{\text{total}} = \lambda_{\text{MSE}}\mathcal{L}_{\text{MSE}} + \lambda_{\text{dep}}\mathcal{L}_{\text{dep}}$, where $\lambda_{\text{MSE}}$ and $\lambda_{\text{dep}}$ are positive hyperparameters. We verify this empirically by randomly setting the parameters of the data generation process and plotting the predictive errors $J_1^*$ and $J_2^*$ for different values of $\beta$. We calculate $J_1^*$ and $J_2^*$ for 5000 runs (shown using thin curves) and plot the average error (shown using thick curves) in Fig. 13. We observe that the average value of $J_1^*$ is always higher than that of $J_2^*$ for all values of $\beta$. But, when $\beta \to 0$, their average values get closer to each other.

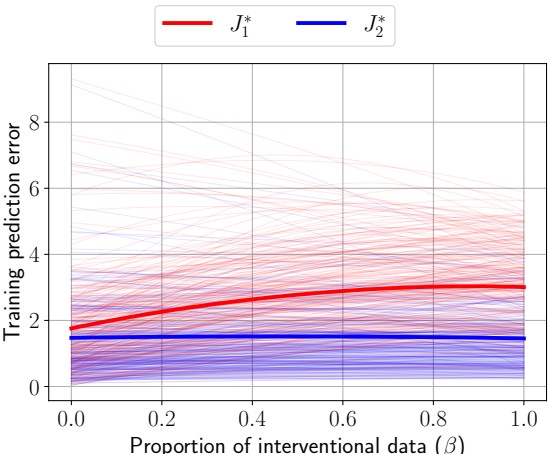

Figure 13: Comparing $J_1^*$ (Eq. (14)) and $J_2^*$ (Eq. (15)) as functions of $\beta$ for 5000 runs with randomly sampled data generation parameters. We show individual runs using thin curves and the average error values using thick curves. We only show the errors from a few randomly sampled runs for visual clarity. We observe that the average value of $J_1^*$ (shown using thick red curve) is always higher than that of $J_2^*$ (shown using thick blue curve), indicating that enforcing linear independence between interventional features is more likely to obtain robust feature extractors than degenerate solutions.

## C   Review of identifiable causal representation learning

The primary objective of identifiable causal representation learning (ICRL) is to learn a representation such that it is possible to identify the latent factors (up to permutation and elementwise transformation) from the representation. These methods are commonly built upon autoencoder-based approaches and learn generative representations. The advantage of learning a causal representation is that the decoder then implicitly acts as the true underlying causal model, facilitating counterfactual evaluation and interpretable representations.

Locatello et al. (2019); Khemakhem et al. (2020) showed that disentangled representation learning was impossible without additional assumptions on both the model and the data. Some of the inductive biases that have been proposed since to learn disentangled representations include auxiliary labels (Hyvarinen &

Morioka, 2016; Hyvarinen et al., 2019; Sorrenson et al., 2020; Khemakhem et al., 2020; Lu et al., 2021; Ahuja et al., 2022b; Kong et al., 2022), temporal data (Klindt et al., 2021; Yao et al., 2022; Song et al., 2023), and assumptions on the mixing function (Sorrenson et al., 2020; Yang et al., 2021; Lachapelle et al., 2022; Zheng et al., 2022; Moran et al., 2022).

**Use of interventional data:** Some works also use interventional data as weak supervision for identifiable representation learning (Lippe et al., 2022b; Brehmer et al., 2022; Ahuja et al., 2022a; 2023; Varıcı et al., 2023; von Kügelgen et al., 2023). Lippe et al. (2022b) learns identifiable representations from temporal sequences with possible interventions at any time step. Similar to our setting, they assume the knowledge of the intervention target. They also assume that the intervention on a latent variable at a time step does not affect other latent variables in the same time step. Lippe et al. (2023) relaxes the latter assumption as long as perfect interventions with known targets are available. Von Kügelgen et al. (2021); Zimmermann et al. (2021) showed that self-supervised learning with data augmentations allowed for identifiable representation learning. Brehmer et al. (2022) use pairs of data samples before and after some unknown intervention to learn latent causal models. Ahuja et al. (2022a) learns identifiable representations from sparse perturbations, with identifiability guarantees depending on the sparsity of these perturbations. Sparse perturbations can be treated as a parent class of interventions where the latent is intervened through an external action such as in reinforcement learning. Ahuja et al. (2022b) use interventional data for causal learning for polynomial mixing functions, under some assumptions on the nature of support for non-intervened variables. Varıcı et al. (2024) relaxes the polynomial assumption on the mixing function and proves identifiability when two uncoupled hard interventions per node are available along with observational data. Varıcı et al. (2023) learn identifiable representations from data observed under different interventional distributions with the help of the score function during interventions. von Kügelgen et al. (2023) uses interventional data to learn identifiable representations up to nonlinear scaling. In addition to the above uses of interventional data, a few works (Saengkyongam & Silva, 2020; Saengkyongam et al., 2024; Zhang et al., 2023) have also attempted to predict the effect of unseen joint interventions with the help of observational and atomic interventions under various assumptions on the underlying causal model.

**Difference from our setting:** The general objective in ICRL is to "learn both the true joint distribution over both observed and latent variables" (Khemakhem et al., 2020). In contrast, the objective of our work is to learn representations corresponding to latent variables that are robust against interventional distributional shifts by leveraging *known* interventional independence relations. We pursue this objective in the hope that as large models such as (Radford et al., 2021), (Brown et al., 2020), (Touvron et al., 2023) and (Dehghani et al., 2023) become more ubiquitous, efficient methods to improve these models with minimal amounts of experimentally collected data will be of interest.

## D Additional Results from Experiments

As mentioned in the main paper, our objective is to improve the robustness of representations against interventional distribution shifts. However, this robustness might come at the cost of observational accuracy since it removes spurious information that gives better performance on observational data. In this section, we report the results of the baselines and our methods on WINDMILL, CelebA, and CivilComments datasets.

## E Visualization of Feature Distribution Learned on Windmill dataset

In this section, we compare the feature distributions learned by RepLIn on WINDMILL dataset against all the baselines from Sec. 5.1. The feature distributions are shown in Fig. 14.

## F Generating Windmill Dataset

We provide the exact mathematical formulation of WINDMILL dataset described in Sec. 3.1. We define the following constants:

| Method | $\beta = 0.5$ | $\beta = 0.3$ | $\beta = 0.1$ | $\beta = 0.05$ | $\beta = 0.01$ |
|---|---|---|---|---|---|
| ERM | $93.85 \pm 1.84$ | $98.06 \pm 1.20$ | $99.70 \pm 0.08$ | $99.92 \pm 0.02$ | $99.98 \pm 0.01$ |
| ERM-Resampled | $94.53 \pm 0.89$ | $94.13 \pm 1.19$ | $94.84 \pm 0.92$ | $94.56 \pm 0.71$ | $94.53 \pm 1.14$ |
| IRMv1 | $93.37 \pm 0.85$ | $93.59 \pm 0.32$ | $93.72 \pm 0.73$ | $92.52 \pm 0.35$ | $94.04 \pm 0.63$ |
| Fish | $95.54 \pm 0.42$ | $95.37 \pm 0.36$ | $95.42 \pm 0.59$ | $95.83 \pm 0.51$ | $96.28 \pm 1.12$ |
| GroupDRO | $82.02 \pm 2.00$ | $84.40 \pm 2.72$ | $85.35 \pm 2.35$ | $84.25 \pm 0.91$ | $92.28 \pm 1.11$ |
| SAGM | $94.77 \pm 0.62$ | $95.17 \pm 0.71$ | $94.13 \pm 1.68$ | $95.61 \pm 0.69$ | $94.04 \pm 1.98$ |
| DiWA | $94.64 \pm 0.96$ | $94.30 \pm 0.36$ | $94.57 \pm 0.64$ | $94.39 \pm 0.99$ | $94.24 \pm 0.59$ |
| TEP | $65.20 \pm 14.22$ | $66.94 \pm 3.78$ | $61.34 \pm 19.35$ | $63.02 \pm 15.59$ | $73.77 \pm 9.01$ |
| RepLIn | $95.16 \pm 0.53$ | $97.83 \pm 0.40$ | $99.24 \pm 0.37$ | $98.75 \pm 0.43$ | $99.10 \pm 0.47$ |
| RepLIn-Resampled | $95.57 \pm 0.62$ | $95.77 \pm 0.68$ | $95.59 \pm 1.08$ | $95.90 \pm 0.35$ | $95.51 \pm 1.71$ |

Table 7: Observational accuracy of various methods used in Sec. 5.1.

| Method | $\beta = 0.5$ | $\beta = 0.3$ | $\beta = 0.1$ | $\beta = 0.05$ | $\beta = 0.01$ |
|---|---|---|---|---|---|
| ERM | $76.87 \pm 1.08$ | $69.86 \pm 3.19$ | $62.78 \pm 1.77$ | $59.52 \pm 1.30$ | $60.15 \pm 3.12$ |
| ERM-Resampled | $73.70 \pm 3.19$ | $71.19 \pm 3.23$ | $73.62 \pm 1.54$ | $71.03 \pm 2.83$ | $70.20 \pm 3.73$ |
| IRMv1 | $78.24 \pm 0.79$ | $74.83 \pm 1.74$ | $78.61 \pm 2.24$ | $76.28 \pm 1.87$ | $71.75 \pm 2.03$ |
| Fish | $77.23 \pm 2.24$ | $77.23 \pm 1.32$ | $78.24 \pm 2.09$ | $76.42 \pm 1.95$ | $73.92 \pm 2.53$ |
| GroupDRO | $80.10 \pm 1.66$ | $80.96 \pm 1.33$ | $80.35 \pm 1.01$ | $77.40 \pm 1.16$ | $71.86 \pm 1.60$ |
| SAGM | $76.43 \pm 2.37$ | $79.05 \pm 2.23$ | $76.96 \pm 4.36$ | $79.86 \pm 1.81$ | $72.81 \pm 3.10$ |
| DiWA | $76.61 \pm 2.15$ | $76.71 \pm 0.59$ | $76.09 \pm 0.69$ | $75.83 \pm 1.83$ | $73.39 \pm 1.31$ |
| TEP | $58.68 \pm 4.72$ | $60.42 \pm 1.30$ | $56.07 \pm 3.35$ | $58.52 \pm 4.36$ | $59.23 \pm 1.13$ |
| RepLIn | $87.94 \pm 1.46$ | $87.76 \pm 2.30$ | $83.23 \pm 2.67$ | $73.63 \pm 2.43$ | $67.52 \pm 2.30$ |
| RepLIn-Resampled | $88.46 \pm 0.96$ | $88.05 \pm 1.04$ | $87.91 \pm 1.36$ | $86.38 \pm 0.85$ | $78.41 \pm 1.27$ |

Table 8: Interventional accuracy of various methods used in Sec. 5.1.

| Method | $\beta = 0.5$ | $\beta = 0.4$ | $\beta = 0.3$ | $\beta = 0.2$ | $\beta = 0.1$ | $\beta = 0.05$ |
|---|---|---|---|---|---|---|
| ERM-Resampled | $91.38 \pm 0.09$ | $91.52 \pm 0.06$ | $91.39 \pm 0.07$ | $90.89 \pm 0.10$ | $90.57 \pm 0.09$ | $91.82 \pm 0.14$ |
| RepLIn-Resampled | $86.02 \pm 0.18$ | $86.35 \pm 0.24$ | $86.58 \pm 0.11$ | $86.94 \pm 0.36$ | $87.67 \pm 0.21$ | $89.83 \pm 0.11$ |

Table 9: Observational accuracy of various methods used in Sec. 5.2.

| Method | $\beta = 0.5$ | $\beta = 0.4$ | $\beta = 0.3$ | $\beta = 0.2$ | $\beta = 0.1$ | $\beta = 0.05$ |
|---|---|---|---|---|---|---|
| ERM-Resampled | $81.09 \pm 0.17$ | $80.56 \pm 0.23$ | $80.06 \pm 0.17$ | $79.08 \pm 0.16$ | $76.63 \pm 0.24$ | $73.42 \pm 0.27$ |
| RepLIn-Resampled | $81.97 \pm 0.14$ | $81.94 \pm 0.17$ | $81.84 \pm 0.18$ | $80.65 \pm 0.22$ | $78.56 \pm 0.20$ | $75.77 \pm 0.05$ |

Table 10: Interventional accuracy of various methods used in Sec. 5.2.

| Method | $\beta = 0.5$ | $\beta = 0.3$ | $\beta = 0.1$ | $\beta = 0.05$ | $\beta = 0.01$ |
|---|---|---|---|---|---|
| ERM-Resampled | $81.26 \pm 0.12$ | $81.77 \pm 0.14$ | $79.78 \pm 0.08$ | $79.97 \pm 0.12$ | $79.13 \pm 0.09$ |
| RepLIn-Resampled | $79.27 \pm 0.09$ | $80.16 \pm 0.12$ | $77.65 \pm 0.06$ | $77.84 \pm 0.12$ | $78.51 \pm 0.16$ |

Table 11: Observational accuracy of various methods used in Sec. 5.3.

$$R_B \sim \mathcal{B}(1, 2.5) \quad\quad\quad \text{(Sample radius)}$$

$$R = \frac{r_{\max}}{2}\left(BR_B + (1 - B)(2 - R_B)\right) \quad\quad \text{(Modify sampled radius based on } B\text{)}$$

$$\Theta_A \sim \mathcal{C}\left(\left\{2\pi\frac{i}{n_{\mathrm{arms}} + 1} : i = 0, \dots, n_{\mathrm{arms}} - 1\right\}\right) \quad\quad \text{(Choose an arm)}$$

$$U \sim \mathcal{U}(0, 1) \quad\quad\quad \text{(To choose a random angle)}$$

29

$$\Theta_{\mathrm{off}} = \theta_{\text{max-off}}\sin\left(\pi\lambda_{\mathrm{off}}\frac{R}{r_{\max}}\right) \quad\quad \text{(Calculate radial offset for the angle)}$$

| Method | $\beta = 0.5$ | $\beta = 0.3$ | $\beta = 0.1$ | $\beta = 0.05$ | $\beta = 0.01$ |
|---|---|---|---|---|---|
| ERM-Resampled | $74.51 \pm 0.07$ | $75.29 \pm 0.22$ | $72.03 \pm 0.18$ | $71.78 \pm 0.12$ | $69.80 \pm 0.45$ |
| RepLIn-Resampled | $75.30 \pm 0.37$ | $75.81 \pm 0.31$ | $72.00 \pm 0.23$ | $71.70 \pm 0.14$ | $69.99 \pm 0.80$ |

Table 12: Interventional accuracy of various methods used in Sec. 5.3.

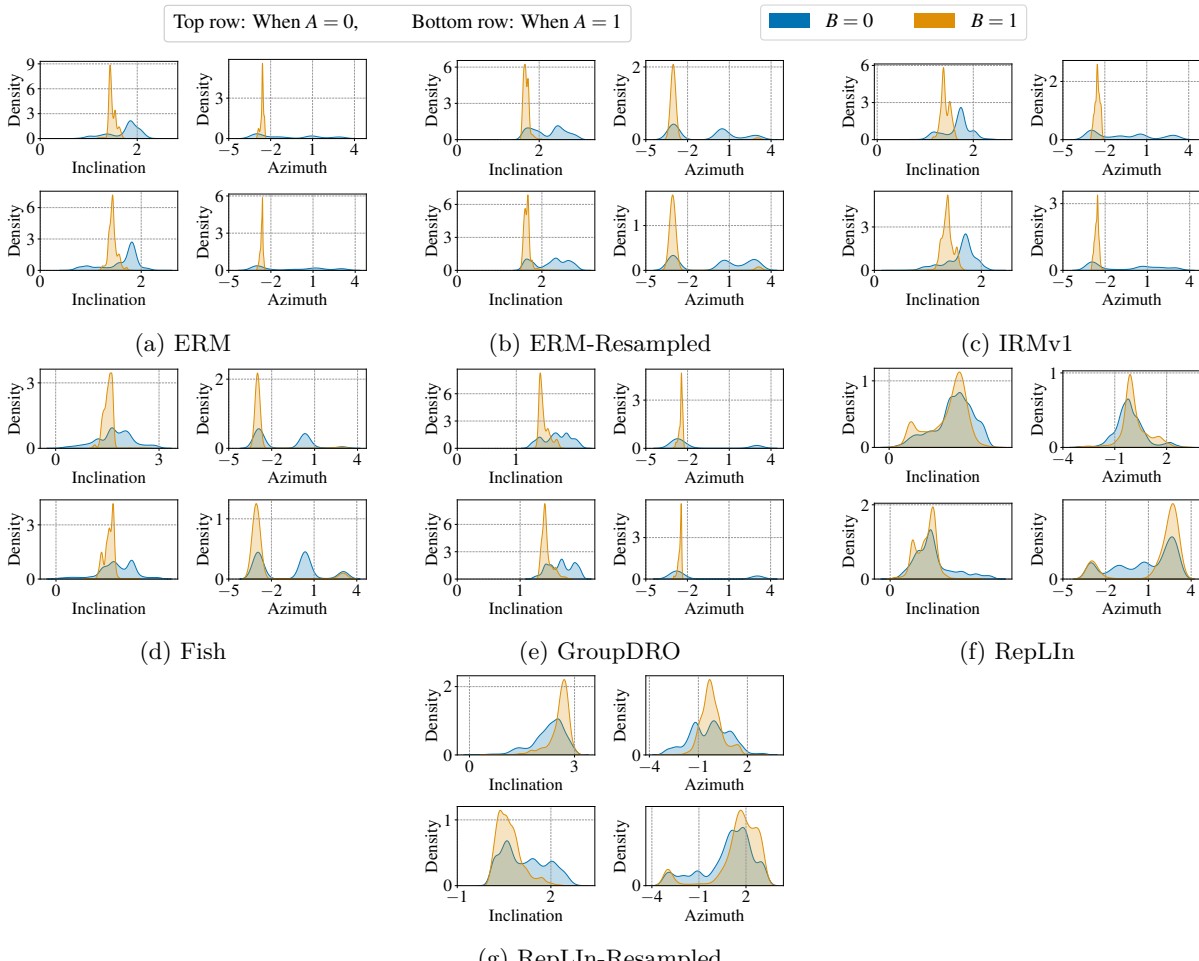

Figure 14: Visualization of interventional features learned by various methods on WINDMILL dataset.

| Constants | Description | Default value |
|---|---|---|
| $n_{\text{arms}}$ | Number of "arms" in WINDMILL dataset | 4 |
| $r_{\text{max}}$ | Radius of the circular region spanned by the observed data | 2 |
| $\theta_{\text{wid}}$ | Angular width of each arm | $\frac{0.9\pi}{n_{\text{arms}}} = 0.7068$ |
| $\lambda_{\text{off}}$ | Offset wavelength. Determines the complexity of the dataset | 6 |
| $\theta_{\text{max-off}}$ | Maximum offset for the angle | $\pi/6$ |

Table 13: Constants used for generating WINDMILL dataset, their meaning, and their values.

PyTorch code to generate WINDMILL dataset is provided in Listing 1.

Listing 1: Code for WINDMILL dataset

```python
import math
import torch

# Constants
num_arms = 4 # number of blades in the windmill
max_th_offset = 0.5236 # max offset that can be added to the angle for shearing (= pi/6)
r_max = 2 # length of the blade
num_p = 20000 # number of points to be generated
offset_wavelength = 6 # adjusts the complexity of the blade

# Sample latent variables according to the causal graph.
A = torch.bernoulli(torch.ones(num_points) * 0.6)
if observational_data:
    B = A
else:
    B = torch.bernoulli(torch.ones(num_points) * 0.5)

# Convert A, B to X.
th_A0 = torch.linspace(0, 2*math.pi, num_arms+1)[:-1]
th_A1 = torch.linspace(0, 2*math.pi, num_arms+1)[:-1] + math.pi/num_arms
# Choose a random arm for A=0 from possible arms. Likewise for A=1.
th_A0 = th_A0[torch.randint(num_arms, (num_p,))]
th_A1 = th_A1[torch.randint(num_arms, (num_p,))]

# beta distribution with alpha=1, beta=3
beta_dist = torch.distributions.beta.Beta(1, 2.5)

# Sample r according to B. If B=0, sample a small r, else sample a large r.
# r ranges from 0 to r_max
B0_r = beta_dist.sample(torch.Size([num_p])) * r_max/2.
B1_r = r_max - beta_dist.sample(torch.Size([num_p])) * r_max/2.
r = B * B0_r + (1-B) * B1_r

# Sample theta according to A.
# Choose the theta arm according to A and then sample from this arm using a uniform distribution.

# First we will have a cartwheel style.
theta = torch.rand(num_p)*th_wid + th_A0*(1-A) + th_A1*A - th_wid/2.

# Add an offset to theta according to r.
th_offset_mod = torch.sin((r/r_max)*offset_wavelength*math.pi)
th_offset = max_th_offset*th_offset_mod
theta += th_offset

x1 = r*torch.cos(theta)
x2 = r*torch.sin(theta)

data = torch.stack([x1, x2], dim=1)
labels = torch.stack([A, B], dim=1).type(torch.long)
```

