# OpenReview forum: "Incorporating Interventional Independence Improves Robustness against Interventional Distribution Shift"
_TMLR — Rejected by TMLR_

### Review · Reviewer_gcZU · 2024-08-09

**Summary Of Contributions:**

This paper tries to mitigate the challenge of learning robust representations under interventional distribution shifts by proposing an algorithm called RepLIn. The authors identify a correlation between performance disparity and adherence to statistical independence conditions in causal models, and derive conditions under which enforcing this independence improves test-time performance on interventional data. RepLIn is demonstrated to enhance robustness against distribution shifts in both synthetic and real-world datasets compared to considered methods.

**Audience:**

Yes

**Broader Impact Concerns:**

No negative societal impacts.

**Claims And Evidence:**

Yes

**Requested Changes:**

See weakness.

**Strengths And Weaknesses:**

***Strength:***

1. This paper is relatively well-structured and well-written.

2. The theoretical analysis is beneficial in the validity of the paper.

***Weakness:***

1. In Table 1, with re-sampling, the performance of predicting A decreases on observation data and increases on intervention data, leading to a seemingly significant relative drop. It is interesting to know authors' comments.

2. In Figure 3, it can be observed that the correlation strength is different when using different metics for feature dependence. In this case, it is premature to claim "a relative drop in accuracy is **always** accompanied by interventional feature dependence". It is beneficial to use more metrics to show the validity of the finding.

3. NHSIC is in fact CKA [a], which is the key component in the loss design. In addition, the proposal of this method seems largely inspired by [b]. However, such similarity have not been discussed.

4. These is solely on real-world dataset experiment (CelebA). It is insufficient to show the validity and generalizablity of the method.

5. Considered baseline methods are out-dated. Some more recent algorithms in domain generalization should be considered.

[a] Similarity of Neural Network Representations Revisited. ICML, 2019.

[b] Self-supervised learning with kernel dependence maximization, NeurIPS, 2021.

---

### Review · Reviewer_tjxF · 2024-08-13

**Summary Of Contributions:**

This paper considers the problem of inferring models from data (which are drawn from some known underlying causal graph) under interventional distribution shift. The paper shows that during interventions on a child of a label node, the joint distribution between the label node, its children and the dataset features shifts. As a result, the test accuracy on the interventional dataset drops on the label node due to dependence between representations with its children in the causal DAG.

The paper shows theoretically (in a linear setting) that the optimal ERM model is not robust to interventional distribution shift, and that by minimizing dependence between representations of the label node and its children, robust models can be found, given a sufficient amount of interventional data. The paper then introduces an algorithm to enforce independence between learned representations and shows that this method minimizes the drop in accuracy under interventional distribution shift.

**Audience:**

Yes

**Broader Impact Concerns:**

There is not much discussion on the limitations of the method, or of its potential use cases / misuses. The paper could benefit from a short section addressing this in the conclusion.

**Claims And Evidence:**

Yes

**Requested Changes:**

In the results section, while I understand that the relative accuracy drop is the main metric of interest, please do include accuracies as presented in Table 1 (on observational and interventional data for A and B separately), at least in an appendix, and please make a mention of it in the main section. It would be very useful to know the "price" that is paid in terms of accuracy on the observational data in order to get robustness to interventional distribution shifts.

For minor suggestions, please see above.

**Strengths And Weaknesses:**

The paper is extremely well-written. Both theoretical and empirical arguments are laid out very nicely, including all mathematical notation, figures and tables. The main ideas of the paper are conveyed clearly and are well-supported through theoretical and empirical arguments.

I did not find any significant weaknesses.

A few suggestions for improving clarity:
1. In the introduction, it is not obvious when it might be useful to have models that are robust under interventional distribution shifts. Typically, one thinks of interventions as being useful for identifying/discovering causal structure. Could the authors supply an example to explain why we might want to build robust classifiers while making interventions on a graph with known causal structure? Are the authors thinking of "interventional distributions" in a broader sense, to include generalization to non-training data from different contexts? Providing an example along with paragraph 2 of the introduction would be particularly helpful.
2. In Section 3, under notation, the term "discriminative" is used repeatedly, but it is unclear at that stage which variables correspond to the features and which to the labels  --- it only becomes clear under _Learning Task_ in Section 3.1. Please do mention this as early as possible.
3. Typo in Section 3.4, under _Statistical Risk_: $\phi^{(A)}$ should be $\Theta^{(A)}$.

---

### Review · Reviewer_2xGU · 2024-09-02

**Summary Of Contributions:**

The paper aims to learn robust representations under distribution shifts by minimizing the dependence between interventional features. The authors empirically demonstrate a correlation between accuracy drops and interventional feature dependence. In a linear setting, they theoretically explain why the optimal ERM model lacks robustness against interventions and how enforcing independence can address this issue. These claims are validated through experiments on both synthetic and real-world datasets.

**Audience:**

Yes

**Broader Impact Concerns:**

No negative social impacts

**Claims And Evidence:**

Yes

**Requested Changes:**

Please see the weakness.

**Strengths And Weaknesses:**

Strength:
1. The paper is well-written and well-organized.
2. The paper provides a theoretical view of representation robustness in a linear setting.
3. The theoretical claims are supported by the empirical evidence.

------------------
Weakness:
1. If I understand correctly, the paper seems to assume the causal structure is already known. This is totally different with the objective of causal representation learning and much simpler compared with it. Please highlight the difference. If I misuderstand and the paper doesn't necessarily require the prior knowledge of causal structure. Plase highlight: first, how to determine the number of latent variables? Second, how to dertmine which is A and which is B, i.e., the causal structure of latent variables?
2. If the causal structure is known and we have access to both data before intervention and after intervention, and we even know the node being intervened. Why not train separate classifiers for data before and after intervention and directly enforcing the independence of features after intervention, such like a distribution specific classifier?
3. A single real-world dataset is not enough to validate the effectiveness of the proposed method and more comparisons will be appreciated.

---

### Decision · Action_Editor_Ni1g · 2024-10-17

**Recommendation:** Reject

**Comment:**

All reviewers acknowledge the theoretical analysis of the paper. However, the support for some critical claims and results is lacking. Specific concerns include the arbitrary selection of hyperparameters and the insufficient comparison with other OOD methods. Furthermore, the paper's framing of the problem could be better defined, either as an OOD issue or a domain adaptation challenge.

Given the journal's lack of an option for major revisions in this round, we can only choose the option for reject, unfortunately. Note that this decision is absolutely not a discouragement but rather an encouragement for resubmission after substantial revision to address these issues. We believe that the paper can be a strong one when addressing the highlighted concerns.

**Audience:**

While the topic of representation learning is highly relevant to TMLR's audience, the paper in its current form might not meet the expectations for a thorough and rigorous analysis. The issues with empirical support and conceptual clarity could limit the paper's appeal and its ability to contribute effectively to ongoing discussions in the field.

**Claims And Evidence:**

The claims made in the submission are only partially supported by the evidence provided. Reviewer gcZU has highlighted significant concerns, including:
- Narrow Empirical Evaluation: The limited number of datasets and the lack of widely recognized benchmarks for out-of-distribution (OOD) generalization weaken the validity of the proposed method. This is compounded by the absence of tailored comparisons with established OOD baseline methods.
- Hyperparameter Inconsistency: The paper uses different hyperparameters for various datasets without a clear or sufficient explanation for these choices. This inconsistency raises doubts about the method's robustness and applicability across different settings.

Additionally, the emphasis on intervention is not adequately justified. Given the settings described, the problem could either be viewed as a domain adaptation issue with known labels in the target domain or as a two-source domain adaptation challenge, but this is not clearly explored in the paper.

**Resubmission Of Major Revision:**

The authors may consider submitting a major revision at a later time.